# Mutation characteristics and molecular evolution of ovarian metastasis from gastric cancer and potential biomarkers for paclitaxel treatment

Pengfei Yu[1,5], Can Hu[1,5], Guangyu Ding[1,5], Xiaoliang Shi[2], Jingli Xu[1], Yang Cao[1], Xiangliu Chen[1], Wei Wu[3], Qi Xu[4], Jingquan Fang [1], Xingmao Huang[1], Shaohua Yuan[2], Hui Chen[2], Zhizheng Wang [2], Ling Huang[1], Fei Pang[2], Yian Du[1] & Xiangdong Cheng [1] ✉

Ovarian metastasis is one of the major causes of treatment failure in patients with gastric cancer (GC). However, the genomic characteristics of ovarian metastasis in GC remain poorly understood. In this study, we enroll 74 GC patients with ovarian metastasis, with 64 having matched primary and metastatic samples. Here, we show a characterization of the mutation landscape of this disease, alongside an investigation into the molecular heterogeneity and pathway mutation enrichments between synchronous and metachronous metastasis. We classify patients into distinct clonal evolution patterns based on the distribution of mutations in paired samples. Notably, the parallel evolution group exhibits the most favorable prognosis. Additionally, by analyzing the differential response to chemotherapy, we identify potential biomarkers, including *SALL4*, *CCDC105*, and *CLDN18*, for predicting the efficacy of paclitaxel treatment. Furthermore, we validate that *CLDN18* fusion mutations improve tumor response to paclitaxel treatment in GC with ovarian metastasis in vitro and *vivo*.

Gastric cancer (GC) is one of the most common malignant tumors worldwide[1], and metastasis and high recurrence rate are the main reasons for poor prognosis. Ovarian metastasis from GC, including synchronous metastasis and metachronous metastasis after radical surgery, accounts for approximately 5–10% of female GC patients[2]. The prognosis of GC with ovarian metastasis is worse compared to other digestive tract-originated metastatic ovarian tumors, with a median survival time of only 8-14 months[3]. The condition of GC with ovarian metastasis is complex, and there is still no consensus on the treatment of this disease. Currently, the systemic treatment regimens for GC

patients with ovarian metastasis typically comprise chemotherapy involving agents such as 5-fluorocrail (5-FU), platinum, or paclitaxel, but the efficacy is still variable and it is not clear which patients could benefit from these treatments. This is also an important reason for treatment failure in female GC patients. Therefore, a comprehensive and systematic approach involving close collaboration among multiple disciplines is required to develop the most suitable individualized treatment plan for these patients.

GC is a heterogeneous disease with unique genomic and phenotypic features, and individual patients often exhibit distinct genetic

[1]Department of Gastric Surgery, Zhejiang Cancer Hospital, Hangzhou Institute of Medicine (HIM), Chinese Academy of Sciences, Hangzhou, Zhejiang, China. [2]Shanghai OrigiMed Co., Ltd, Shanghai, PR China. [3]Department of Pathology, Zhejiang Cancer Hospital, Hangzhou Institute of Medicine (HIM), Chinese Academy of Sciences, Hangzhou, Zhejiang, China. [4]Department of Oncology, Zhejiang Cancer Hospital, Hangzhou Institute of Medicine (HIM), Chinese Academy of Sciences, Hangzhou, Zhejiang, China. [5]These authors contributed equally: Pengfei Yu, Can Hu, Guangyu Ding. ✉e-mail: abdsurg@163.com

and molecular profiles[4]. The advent of next-generation sequencing (NGS) has rapidly expanded our knowledge of the genetic basis of this disease. Several studies have provided insights into GC from the perspectives of gene mutations and potentially actionable targets[5–7]. The Cancer Genome Atlas (TCGA) investigators have published comprehensive genomic and transcriptomic analyses of GC and categorized GC into four distinct subtypes, characterized by Epstein-Barr viral infection (EBV), microsatellite instability (MSI), chromosomal instability (CIN), and genomic stability (GS)[5]. Recently, several molecular classifications of different GC subtypes have also been proposed[8–10]. Studies related to molecular characteristics have explored potential predictive biomarkers for the prognosis of GC and have guided personalized treatment[11,12]. However, ovarian metastases from GC are often less sensitive to chemotherapy and radiotherapy and lack effective therapeutic targets[13,14]. Thus, it is urgent to elucidate the molecular characteristics and facilitate the development of therapies for this disease.

In this study, we presented the molecular landscape of ovarian metastasis from GC by performing whole-exome sequencing (WES) on the collected primary-metastasis samples. These results illustrated the mutation characteristics of primary gastric and metastatic ovarian lesions, as well as the molecular heterogeneity between synchronous and metachronous ovarian metastases. We also identified potential biomarkers for predicting the efficacy of paclitaxel treatment in GC patients with ovarian metastasis. Our study elucidated mutation characteristics and molecular evolution of ovarian metastasis from GC and provided a rationale for the development of specialized treatment.

## Results

### Patient characterization

A total of 74 GC patients with ovarian metastasis including 53 synchronous and 21 metachronous were enrolled (Fig. 1a). The clinicopathologic characteristics of these patients were summarized in Table 1. The median age of these patients was 46 (range: 28-73) years. Tumors were located in the proximal stomach in 8 (10.8%) patients, in the middle stomach in 40 (54.1%) patients, and in the distal stomach in 26 (35.1%) patients. The majority (74.3%) of these patients had bilateral ovarian metastases, and 49 (66.2%) patients had ovarian metastases combined with peritoneal or other metastases. According to Lauren's classification, 24 patients (32.4%) were diffuse type, 21 patients (28.4%) were intestinal type, and 29 patients (39.2%) were mixed type. Most of the primary tumors (87.8%) were poorly differentiated. Among these patients, 62 (83.8%) were non-signet ring cell adenocarcinomas, 5 (6.7%) were signet ring cell adenocarcinomas, and 7 (9.5%) were unknown subtype (Table 1).

### Molecular heterogeneity of primary gastric lesion and metastatic ovarian lesion

A total of 138 tumor tissues, including 65 primary gastric tumor and 73 ovarian metastatic tumor samples, were collected from the 74 enrolled patients. All samples were tested using WES. In primary gastric lesions, a total of 6,668 genetic alterations (GAs) were identified, including 5,638 (84.6%) substitution/indels, 586 (8.8%) gene amplifications, 399 (6.0%) truncations, 43 (0.6%) fusions/rearrangements, and 2 (0.03%) gene homozygous deletions (Fig. 1b). For metastatic ovarian lesions, we determined 11,409 GAs, including 9,290 (81.4%) substitutions/indels, 1,145 (10.0%) gene amplifications, 902 (7.9%) truncations, 56 (0.5%) fusions/rearrangements, and 16 (0.1%) gene homozygous deletions. Compared to non-signet ring cell adenocarcinoma, a higher frequency of mutations in CCND1 (P = 0.024) was observed in signet ring cell adenocarcinoma (Fig. 1b). Through the comparative analysis of single nucleotide variants (SNVs), we determined a significant correlation between metastatic and primary lesions within individuals, and the

correlation coefficient ranged from 0 to 0.96 (Fig. 2a). The correlation between individual primary and metastatic copy number variations (CNVs) was low (Fig. 2b). Statistical analysis showed that the mutation frequencies for FUS (P = 0.0004), ETV4 (P = 0.0221), CBFB (P = 0.0221), PDGFRB (P = 0.0221), NTRK1 (P = 0.0482), and RAD51 (P = 0.0482) were lower in metastatic ovarian lesions, compared to primary gastric lesions (Fig. 2c). CNVs identified in the primary gastric lesion predominantly comprised gene amplifications and were primarily localized on chromosomes 7, 11, 16, and 17 (Fig. 2d). Notably, apart from the prevalent gene amplifications typically observed on chromosomes 3, 5, 10, 11 and 17, we also detected fragment deletions on chromosome 9 in the metastatic ovarian lesions (Fig. 2d). In this patient cohort, we identified a total of 13 gene fusions in primary gastric tumors and 12 gene fusions in metastatic ovarian tumors. Strikingly, the most prevalent fusion gene observed in both primary and metastatic tumors was CLDN18. Furthermore, we noted identical fusion types in the primary and metastatic tumors of five patients. These included three patients of CLDN18-ARHGAP26 fusion, one patient of CLDN18-ARHGAP42 fusion, and one patient of TCF3-MBD3 fusion (Fig. 2e).

The most frequent point mutation types were C > T/G > A transitions and C > T/G > A transversions (Supplementary Fig. 1). Three distinct single base substitution (SBS) signatures, denoted as SBS1, SBS17b, and SBS6, were identified through rigorous screening, guided by a cosine similarity > 0.85 (Fig. 2f, Supplementary Data 1). SBS1 exhibits a clock-like signature and demonstrates a clear correlation with the age of patients[15]. Meanwhile, SBS17b is potentially linked to the administration of 5-fluorouracil (5-FU) chemotherapy and the oxidative damage induced by reactive oxygen species[16]. Furthermore, SBS6 is notably associated with impaired DNA mismatch repair mechanisms and is predominantly observed in microsatellite unstable tumors[17]. The individual distribution of the three identified mutational features was shown in Fig. 2g. No significant correlation between the mutational features and any of the clinicopathological features was determined (Supplementary Fig. 2).

### Different mutational characteristics between synchronous and metachronous ovarian metastasis

Among the 64 patients with paired primary gastric and metastatic ovarian samples, 45 exhibited synchronous metastasis, while 19 presented with metachronous metastasis. We compared the SNV, CNV, and mutation characteristics of primary lesions and ovarian metastatic lesions derived from synchronous and metachronous patients, respectively. Data analysis shows that both synchronous and metachronous patients exhibit similar mutation patterns in terms of mutation characteristics (Fig. 3a). Compared with synchronous patients in the primary lesion sample, the mutation frequency of PIK3CA was significantly higher in metachronous patients (Fig. 3b). In metastatic lesions, the frequency of ERBB3 mutations is significantly higher in synchronous patients compared to metachronous patients (Fig. 3b). In the primary lesions, synchronous patients exhibited a higher number of CNV variations, which clustered on chromosomes 5, 7, 11, 16, and 17 (Supplementary Fig. 3). In the metastatic lesions, metachronous patients showed a higher number of CNV variations, which clustered on chromosomes 7, 17, and 19 (Supplementary Fig. 3). The proportion of mutations shared between primary and metastatic tumors in synchronous metastatic patients ranged from 0% to 84.8%, while it varied from 0.8% to 73.2% in metachronous metastatic patients (Fig. 3c). We found that patients with synchronous metastases exhibited a relatively lower proportion of specific mutations in the primary GC (P = 0.046). However, there was no significant difference in the proportion of specific mutations in metastatic ovarian lesion and shared mutations between patients with synchronous and metachronous metastases (Fig. 3d).

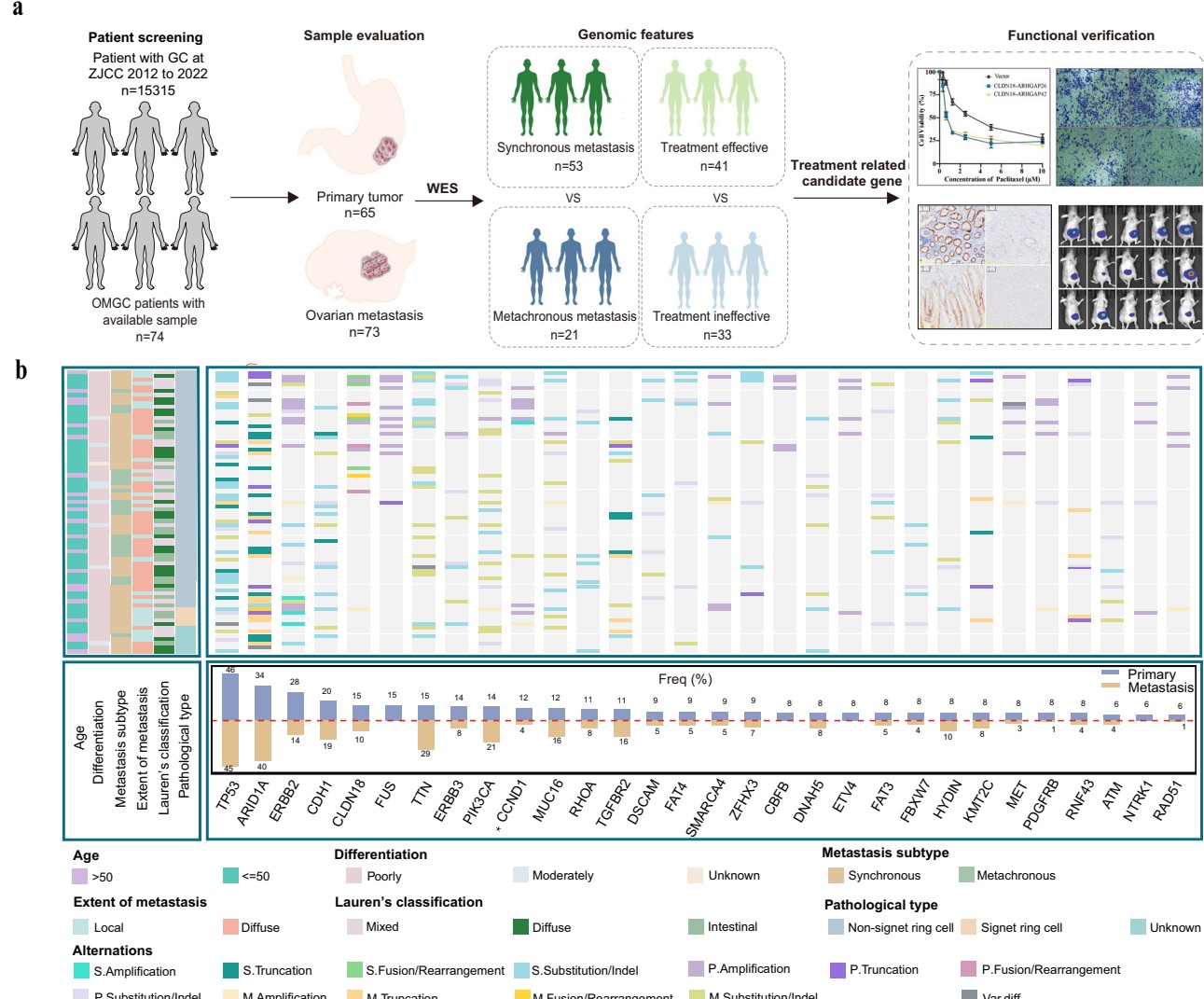

**Fig. 1 | Study protocol flow chart and mutational landscape of our cohort.**
**a** Study protocol flow chart. From a cohort of 74 patients with ovarian metastases from GC, 65 primary tumor samples and 73 metastatic lesion samples were obtained. Analyses were conducted to investigate the correlation between gene characteristics and synchronous or metachronous metastasis groups, as well as the therapeutic efficacy of paclitaxel treatment. Furthermore, immunohistochemistry and in vitro and *vivo* functional studies validation were performed to assess the expression and functional relevance of potential gene biomarkers associated with treatment response. ZJCC, Zhejiang Cancer Center. OMGC, ovarian metastases from gastric cancer. **b** The mutational landscape of primary GC and metastatic ovarian cancer. The middle panel shows somatic gene alterations by patient (row) and by gene (column) (The prefix S means shared alterations, the prefix M means alterations only in metastasis, and the prefix P means alterations only in primary). The histogram on the bottom shows the number of alterations accumulated on top 30 listed genes in each individual sample. The trajectory on the left displays histopathological features, such as age, differentiation, pathological type, metastasis subtype, extent of metastasis and Lauren's classification. Freq, frequency. S., same. P. primary. M., metastasis. Var.diff, the variation is different between primary and metastatic lesions.

## Integrated pathway analysis of primary gastric lesion and metastatic ovarian lesion and between synchronous and metachronous ovarian metastasis

After classifying all primary gastric and metastatic ovarian samples by metastasis subtype, we further integrated mutation data of five groups of samples (primary, metastatic, synchronous, metachronous and TGGA database) for the enrichment analysis of signaling pathways in high-frequency mutated genes (Supplementary Data 2). Subsequently, for signaling pathways with FDR < 0.05, we calculated scores based on gene mutations and their frequencies within each pathway and performed clustering, ultimately presenting the signaling pathway status between different groups in Fig. 4a. The results show that the p53, thyroid hormone, neurotrophin, sphingolipid, ErbB, Wnt, FoxO, AGE-RAGE, Apelin, PI3K-Akt and MAPK signaling pathways consistently achieved elevated scores across all cohorts (Fig. 4a). After excluding some genes and pathways with high mutation redundancy, we focused on signaling pathways of p53, ErbB, Wnt, PI3K-Akt, and MAPK, which were associated with cell proliferation, migration, and differentiation (Fig. 4b). In the ErbB and MAPK signaling cascades, the mutational spectrum was mainly enriched in the upstream effector genes, including but not limited to *EGFR*, *ERBB2*, and *ERBB3*. The Wnt signaling pathway exhibited alterations in a series of regulatory factors, particularly involving *CTNNB1*, *APC*, *AXIN1* and *RNF43*. In the p53 and PI3K-Akt pathways, mutations were concentrated at key regulatory sites, such as *TP53*, *CDKN2A*, *CDKN2B*, *PIK3CA*, and *PTEN*, highlighting their critical roles in pathway regulation (Fig. 4b).

**Table 1 | The clinicopathologic features of gastric cancer patients with ovarian metastasis**

| Total | | n = 74 |
|---|---|---|
| **Metastasis subtype (n/%)** | Synchronous | 53 (71.6%) |
| | Metachronous | 21 (28.4%) |
| **Age(years)** | Median (range) | 46 (28-73) |
| **Tumor location (n/%)** | Proximal stomach | 8 (10.8%) |
| | Middle stomach | 40 (54.1%) |
| | Distal stomach | 26 (35.1%) |
| **Lauren's classification** | Diffuse | 24 (32.4%) |
| | Intestinal | 21 (28.4%) |
| | Mixed | 29 (39.2%) |
| **Laterality (n/%)** | Bilateral | 55 (74.3%) |
| | Unilateral | 17 (23%) |
| | Unknown | 2 (2.7%) |
| **Extent of metastasis (n/%)[a]** | Local metastasis | 25 (33.8%) |
| | Diffuse metastasis | 49 (66.2%) |
| **Differentiation (n/%)** | High | 0 (0) |
| | Moderately | 8 (10.8%) |
| | Poorly | 65 (87.8%) |
| | Unknown | 1 (1.4%) |
| **Pathological type (n/%)** | Non-signet ring cell | 62 (83.8%) |
| | Signet ring cell | 5 (6.7%) |
| | Unknown | 7 (9.5%) |
| **T stage (n/%)** | T1 | 0 (0%) |
| | T2 | 6 (8.1%) |
| | T3 | 10 (13.5%) |
| | T4 | 36 (48.7%) |
| | Unknown | 22 (29.7%) |
| **N stage (n/%)** | N0 | 3 (4.1%) |
| | N1 | 5 (6.8%) |
| | N2 | 9 (12.2%) |
| | N3 | 29 (39.2%) |
| | Unknown | 28 (37.8%) |
| **Samples composition (n/%)** | Gastric lesion only | 1 (1.4%) |
| | Ovarian lesion only | 9 (12.2%) |
| | Gastric lesion and ovarian lesion | 64 (86.5%) |

[a]Local metastasis: ovarian metastasis only; Diffuse metastasis: with extraovarian metastasis.

## Complex genetic evolution showed the heterogeneity of ovarian metastasis from GC

The median proportion of shared mutations between primary gastric and metastatic ovarian tumors was 24.9%, ranging from 0.0% to 84.8% (Supplementary Data 3). Notably, two out of the 64 patients exhibited no shared mutations between their paired samples. The most frequently occurring mutated genes observed in shared mutations were TP53 (22/64, 34.38%), ARID1A (14/64, 21.88%), CDH1 (11/64, 17.19%), TTN (8/64, 12.50%), ERBB2 (7/64, 10.94%), and TGFBR2 (7/64, 10.94%).

To survey the genetic evolution of these patients, such as parallel evolution and linear evolution[18], we reconstructed the phylogenetic tree and calculated the genomic distance among primary and metastatic tumors in each patient (Supplementary Data 3). Among the 55 patients with a well-defined evolutionary relationship were classified into three distinct migration patterns, including parallel evolution, linear evolution, and intermediate evolution (Fig. 5a). There was a few shared GAs in the paired samples of 18 patients, with a median genomic distance of 8.0% and a range of 0.8% to 29.3%, which was defined as parallel evolution. The parallel evolution shared a short trunk, and the

phylogenetic tree involved multiple branches from a founder clone, suggesting that tumor cells from metastatic and primary tumors of these patients developed independently at an early stage. The paired samples of 20 patients had more shared mutations, with a median genomic distance of 62.7% and a range of 25.6% to 84.8%, which was defined as linear evolution. The linear evolution shared a long trunk that harbored the initial cancer driver alteration, indicating that a founder clone that acquired driver alteration disseminated late from the primary tumor and evolved into the metastatic tumor. Seventeen patients with a median number of mutations between the trunk and branches were considered as intermediate evolution (Fig. 5a). In addition, the proportion of linear evolution in the metachronous group was higher than that in the synchronous group (50.0% vs 29.7%, P = 0.232), while the proportion of parallel evolution in the synchronous group was higher compared to the metachronous group (40.5% vs 16.7%, P = 0.050) (Supplementary Fig. 4).

The distinct phylogenetic patterns and genetic similarities observed in metastatic GC may have implications for clinical outcomes. Therefore, we performed an efficacy assessment and conducted survival analysis on patients exhibiting varying evolutionary patterns. Following the categorization of patients and the exclusion of samples with an insufficient number of mutations, which are not conducive to evolutionary analysis, we found that the proportion of patients who respond to paclitaxel in parallel evolution group was higher than that in linear group (72.2% vs 40.0%, P = 0.059), while there was no significant difference between parallel evolution group and intermediate group (72.2% vs 64.7%, P = 1) (Fig. 5b). Survival analysis confirmed that patients in parallel evolution group had a better prognosis than those in the linear group (5-year OS: 24.93% vs 0%, P = 0.029), while there was no significant difference in OS between parallel evolution group and intermediate group (5-year OS: 24.93% vs 30.20%, P = 0.981) (Fig. 5c).

## Exploration of potential factors affecting the efficacy of paclitaxel

The systemic treatment regimen for GC patients with ovarian metastasis typically comprises chemotherapy involving agents such as 5-FU, platinum, or paclitaxel. Our previous research has found that paclitaxel is effective for the ovarian metastasis from GC[19]. In this cohort, 41 patients responded to paclitaxel were considered as the "effective" group, and 33 patients did not respond to paclitaxel were considered as the "ineffective" group (supplementary Fig. 5). We conducted an investigation into the clinical factors associated with the response to paclitaxel treatment. Our findings revealed that patients with metachronous ovarian metastasis exhibited a higher response rate compared to those with synchronous ovarian metastasis (77.8% vs 48.7%, P = 0.047). Additionally, patients with local metastasis displayed a higher response rate than those with diffuse metastasis (83.3% vs 39.4%, P = 0.002) (Supplementary Fig. 6).

Then, we also explored gene mutations related to the efficacy of paclitaxel. The frequency of variation of seven genes tended to be different between the effective group and ineffective groups. Statistical analysis indicated that the mutations of RHOA (P = 0.037), AFF2 (P = 0.028), PIK3CD (P = 0.028), and TAF1L (P = 0.028) were associated with the ineffectiveness of paclitaxel treatment, while the mutations of SALL4 (P = 0.036), CCDC105 (P = 0.018), and CLDN18 (P = 0.036) were associated with the response to paclitaxel treatment (Fig. 6a). Mutations of AFF2, PIK3CD, and TAF1L were specifically detected in patients within the ineffective group, while mutations of CCDC105, CLDN18, and SALL4 were specifically detected in patients within the effective group. Exception for the gene rearrangement of CLDN18- ARHGAP26/42 (Fig. 6b), most of the mutation types of these genes were substitutions/indels.

In addition, we conducted a analysis of these efficacy-related genes in metastatic ovarian lesions and their corresponding primary

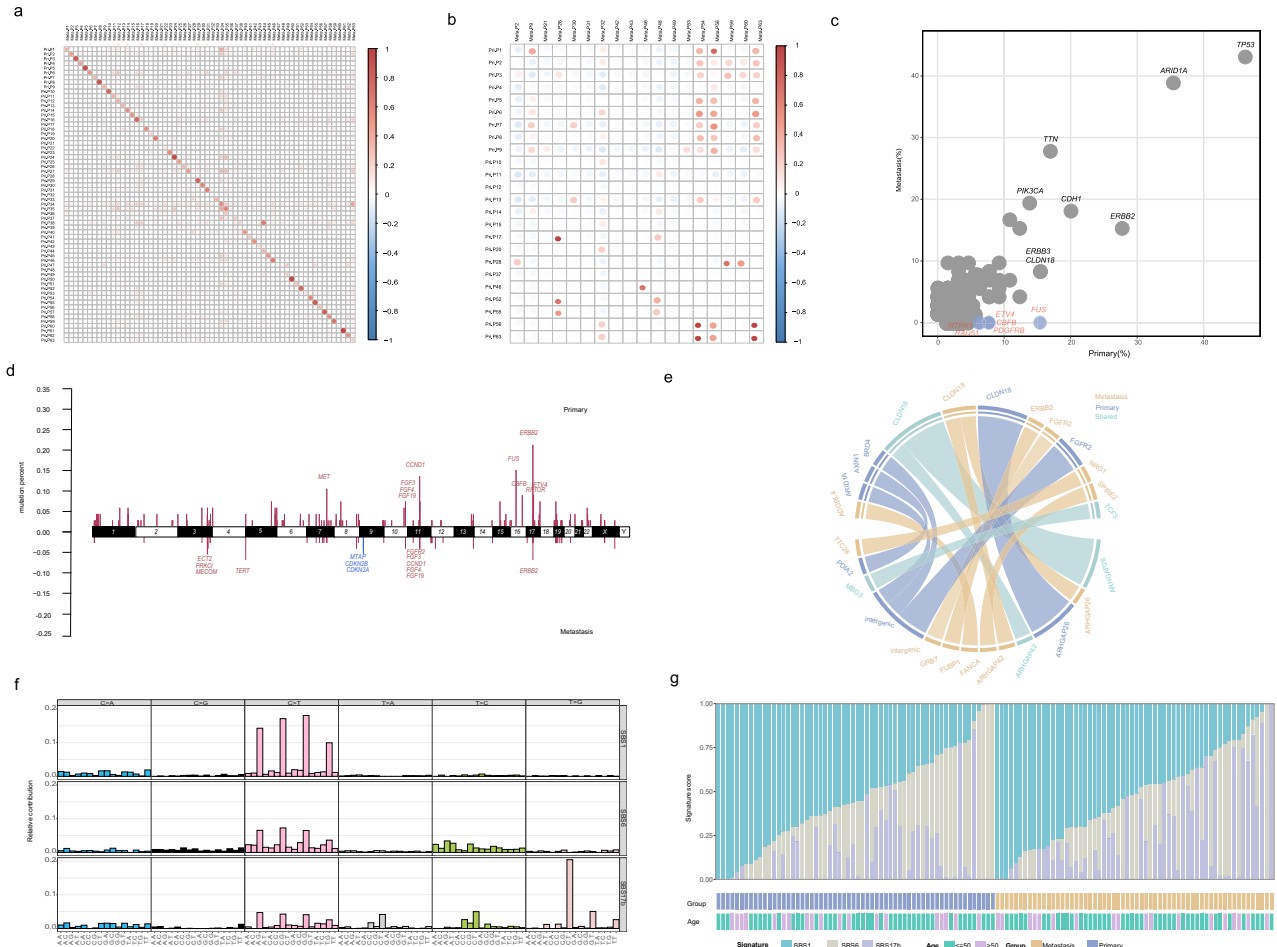

**Fig. 2 | Comparison of mutation characteristics between primary and metastatic lesions in patients with ovarian metastasis of gastric cancer. a** A pearson's correlation analysis of individual SNVs in primary gastric and metastatic ovarian lesions. Pri, primary. Meta, metastasis. **b** A pearson's correlation analysis of individual CNVs in primary gastric and metastatic ovarian lesions. Pri, primary. Meta, metastasis. **c** The comparison of alterations in primary and metastatic ovarian lesions. Chi-squared test ($\chi^2$) and Fisher's exact test were used in the comparison. **d** Comparison of chromosome distribution of CNVs in primary gastric and metastatic ovarian lesions. The upper half (primary) represents the CNV distribution in primary gastric lesions and the lower half (metastasis) represents the CNV

distribution in metastatic ovarian lesions. Chi-squared test ($\chi^2$) and Fisher's exact test were used in the comparison. **e** A comparison of fusion events in primary gastric and metastatic ovarian lesions. Blue indicates specific fusions in primary lesions, orange indicates specific fusions in metastatic lesions, and cyan indicates shared fusions. **f** Ninety-six substitutions were derived from WES data obtained from 64 pairs of primary gastric and metastasis ovarian tumor samples. The horizontal axis represents the mutation patterns for 96 substitutions using different colors. The vertical axis depicts estimated mutations attributed to a specific mutation type. **g** The distribution of mutation signatures in the cohort. SBS, single base substitution.

gastric lesions. Based on their distribution patterns, these genes can be classified into three subtypes (Supplementary Fig. 7). The first subtype encompasses mutations that consistently present in both primary gastric lesions and metastatic tumors, such as *RHOA* (6 mutations in primary gastric lesions/6 mutations in metastasis tumors) and *TAF1L* (1/1). The second subtype includes mutations that are partially present in primary gastric lesions, such as *SALL4* (2/3), *CCDC105* (1/3), *CLDN18* (4/7), and *PIK3CD* (1/3). And the third subtype is defined by mutations are absent in primary gastric lesions and exist exclusively in the metastatic lesions, such as *AFF2* mutation (0/1). We further analyzed the relationships between these mutations and the prognosis of patients. The results showed that *CCDC105* mutations tended to be associated with good prognosis, while *RHOA*, and *PIK3CD* mutations tended to be associated with poor prognosis (Supplementary Fig. 8).

**The effectiveness of chemotherapy (paclitaxel) is associated with CLDN18 fusion**

Previous studies have shown that CLDN18 fusion is associated with poor response to 5-Fu/oxaliplatin chemotherapy in gastric signet ring cell carcinoma[20,21]. Interestingly, we found that all patients with

CLDN18 fusion are sensitive to paclitaxel treatment. These remind us that CLDN18 fusion may be an important marker to predict the efficacy of chemotherapy in GC. To further clarify the correlation between CLDN18 fusion and chemotherapy response, we constructed a stable human MKN-1 and HGC-27 GC cell lines by transfecting CLDN18-ARHGAP26/42 lentivirus. The CCK-8 assay confirmed that CLDN18-ARHGAP26/42 fusion can promote the ability of paclitaxel to inhibit the proliferation of GC cells, while these fusions had no significant effect on oxaliplatin inhibition of GC cell proliferation (Fig. 6c). The transwell experiments proved that CLDN18-ARHGAP26/42 fusion can promote the ability of paclitaxel to inhibit the invasion and metastasis of GC cells, while these fusions had no significant effect on the invasion and migration ability of GC cells (Fig. 6d, e). Furthermore, we assessed the function of CLDN18-ARHGAP26/42 fusion in GC ovarian metastasis mouse model. MKN-1 GC cells stably transfected with *CLDN18-ARHGAP26/42* fusion mutations or empty vector were subcutaneously inoculated into left ovary of nude mice. One week later, mice were treated with 10 mg/kg/tiw paclitaxel for 4 weeks. The results showed that the CLDN18-ARHGAP26/42 fusion significantly increased the sensitivity of ovarian metastasis in GC to paclitaxel (Fig. 6f–h).

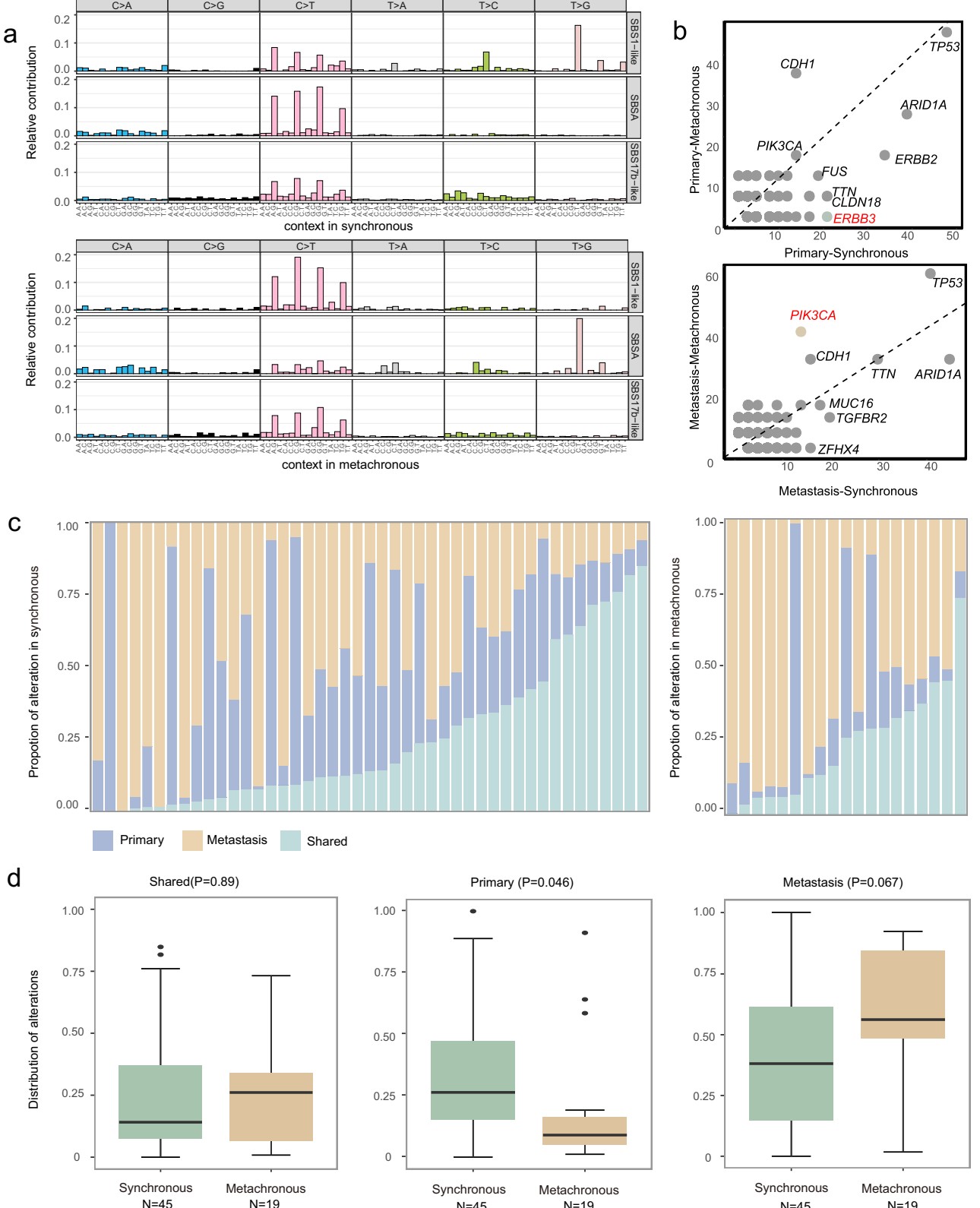

**Fig. 3 | Comparative analysis of mutation characteristics between synchronous and metachronous ovarian metastasis. a** Ninety-six substitutions were derived from WES data obtained from synchronous and metachronous ovarian metastasis tumor samples. SBS, single base substitution. **b** The enrichment of alterations in primary or metastatic ovarian lesions between synchronous and metachronous ovarian metastasis. Chi-squared test ($\chi^2$) and Fisher's exact test were used in the comparison. **c** The distribution of primary, metastatic, and shared genetic changes in patients with synchronous and metachronous ovarian metastasis. **d** Comparative analysis of mutation composition in synchronous ($n = 45$) and metachronous ovarian metastasis ($n = 19$). *P*-values were calculated using a one-way ANOVA analysis of variance with two-side. The boxplot elements indicate the maxima, 75th percentile, median, 25th percentile, and minima. Source data are provided as a Source Data file.

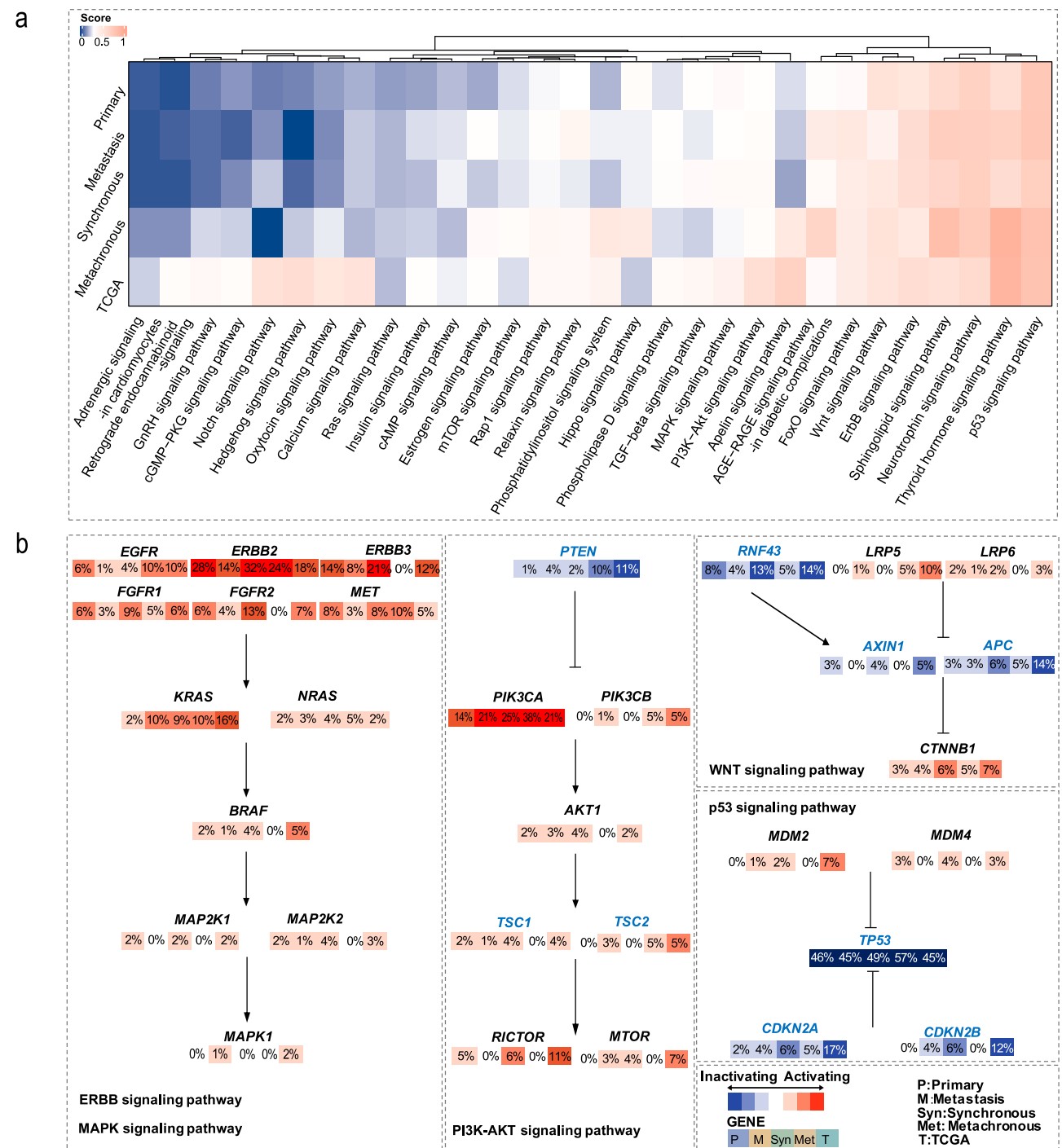

**Fig. 4 | Genomic alterations in signaling pathways in primary gastric lesions, ovarian metastasis and between synchronous and metachronous ovarian metastasis. a** The heatmap shows the enrichment of KEGG pathway mutations in the four groups of our cohort in comparison to TCGA GC. Scores ranging from 0 to 0.25 were colored to varying degrees in blue based on their size, scores 0.25 were marked as white, and scores ranging from 0.25 to 1 were marked as varying degrees in red. **b** The distribution of mutational frequencies in the ErbB, MAPK, PI3K-Akt, Wnt and p53 signaling pathways across four groups and TCGA database. Red indicates predicted activation, blue indicates predicted inactivation. The numerical value superimposed on each box corresponds the frequency of gene mutations in the corresponding groups. The intensity of the color saturation within each box is directly proportional to the mutational frequency. P., primary. M., metastasis. Syn., synchronous. Meta., metachronous. T., TCGA.

In recent years, CLDN18 has become an important target for the treatment of GC[22,23]. We further examine the relationship between CLDN18 expression, CLDN18 fusion and paclitaxel sensitivity. The CLDN18 expression of primary gastric tumor, ovarian metastatic tumor and adjacent paracancerous tissues of these patients was detected via immunohistochemistry (IHC, Supplementary Data 4). The results showed that the CLDN18 expression of primary gastric tumor was positive in 52.9% (18/34) patients in the "effective" group and 54.2% (13/24) in the "ineffective" group ($\chi^2 = 0.071$, $P = 0.503$, Supplementary Fig. 9a), while the CLDN18 expression of metastatic tumor was positive in 31.58% (12/38) patients in the "effective" group and 30.43% (7/23) in the "ineffective" group ($\chi2 = 0.009$, $P = 0.925$, Supplementary Fig. 9b).

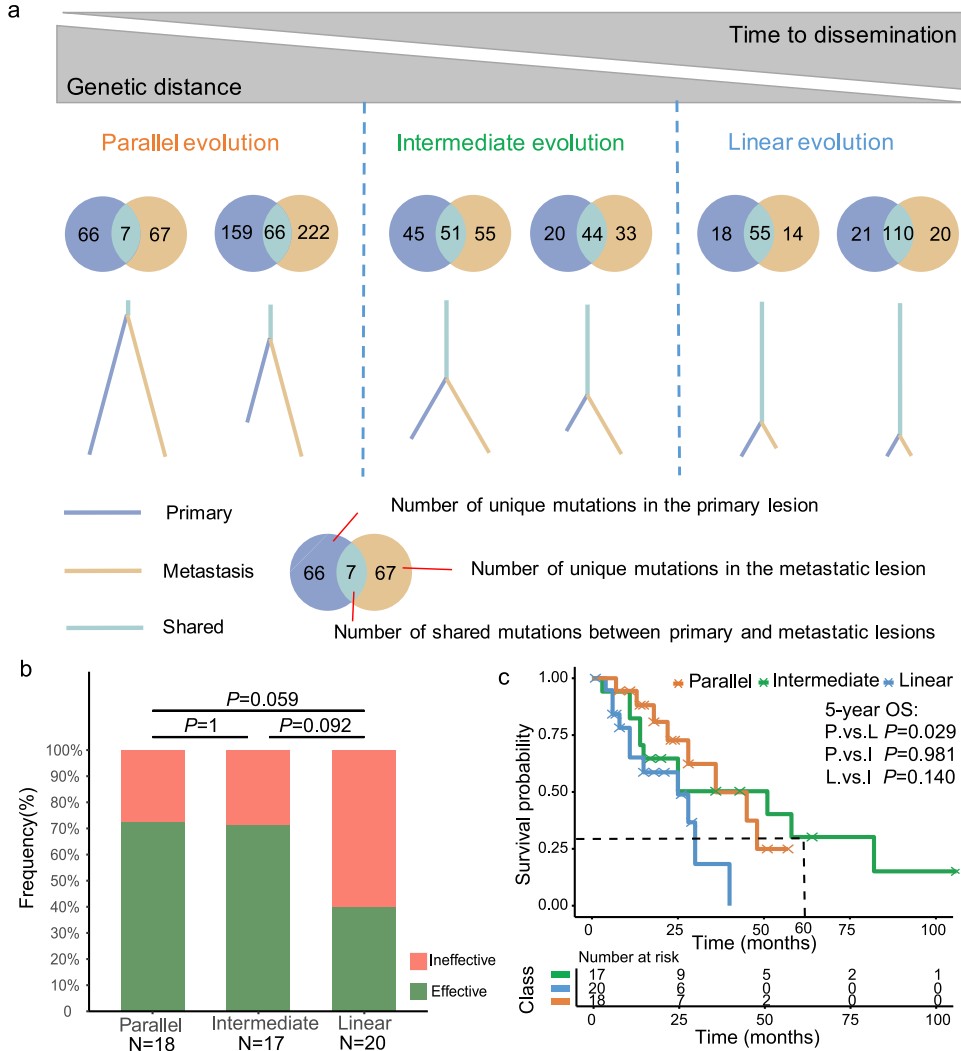

**Fig. 5 | Inferred phylogeny and migration patterns of ovarian metastatic GC with genomic similarity and the resulting prognosis. a** Phylogenetic relationships in paired primary tumors and metastatic tumors. The phylogenetic intermediate trunk represents shared mutations between primary and metastatic lesions, left branches represent unique mutations in primary lesions, and right branches represent unique mutations in metastatic lesions. **b** Response of ovarian lesions with different evolutionary patterns to paclitaxel treatment. *P*-values were calculated using Fisher's exact test between any two cohorts. Source data are provided as a Source Data file. **c** Kaplan-Meier's curves for overall survival based on patients with different evolutionary patterns after diagnosis. P, parallel evolution. I, intermediate evolution. L, linear evolution. Differences between groups were assessed by the log-rank test. *P* value < 0.05 was considered statistically significant.

What is more, there was no significant relationship between CLDN18 fusion and CLDN18 expression in ovarian metastasis (4/6 vs 15/55, *P* = 0.130, Supplementary Fig. 10).

## Discussion

Ovarian metastasis is a prevalent occurrence in female GC patients and often exhibits limited responsiveness to chemotherapy[24], substantially impacting patient prognosis. This investigation delves into the distinct mutational profiles of ovarian metastases and their corresponding primary gastric tumors through WES. It dissects dissimilarities in gene mutations and tumor signaling pathways between patients with primary tumor and ovarian metastases, synchronous and metachronous metastases in GC patients with ovarian metastasis. Furthermore, this study uncovers various clonal evolution patterns within GC patients suffering from synchronous and metachronous ovarian metastases, possibly contributing to the observed heterogeneity. We also conducted a focused analysis to identify potential biomarkers for predicting paclitaxel sensitivity. In essence, our research highlights the heterogeneity, diversity, and complexity associated with ovarian metastasis originating from GC, offering valuable insights into

potential biomarkers for forecasting sensitivity to paclitaxel-based chemotherapy.

Tumor heterogeneity is a fundamental hallmark of malignancies, exerting profound influences on various aspects of cancer, including tumorigenesis, evolution, metastasis, and therapeutic responses. Intratumor heterogeneity, in particular, serves as the driving force behind tumor evolution and the development of resistance to treatments[21]. Our study revealed the consistent rates of different mutation types in primary gastric and ovarian metastases, indicating that SNVs are more likely to spread than CNVs during tumor evolution. Although there is some similarity between primary gastric tumors and ovarian metastases after enrichment through signaling pathways, there is still significant heterogeneity between primary tumors and ovarian metastases in different patients. These molecular-level inconsistencies correlate with divergent trends in tumor evolution among GC patients with ovarian metastasis. Such outcomes underscore the considerable genetic diversity present in primary gastric and metastatic ovarian tumors across distinct patients, shedding light on the intricate processes involved in ovarian metastasis arising from GC.

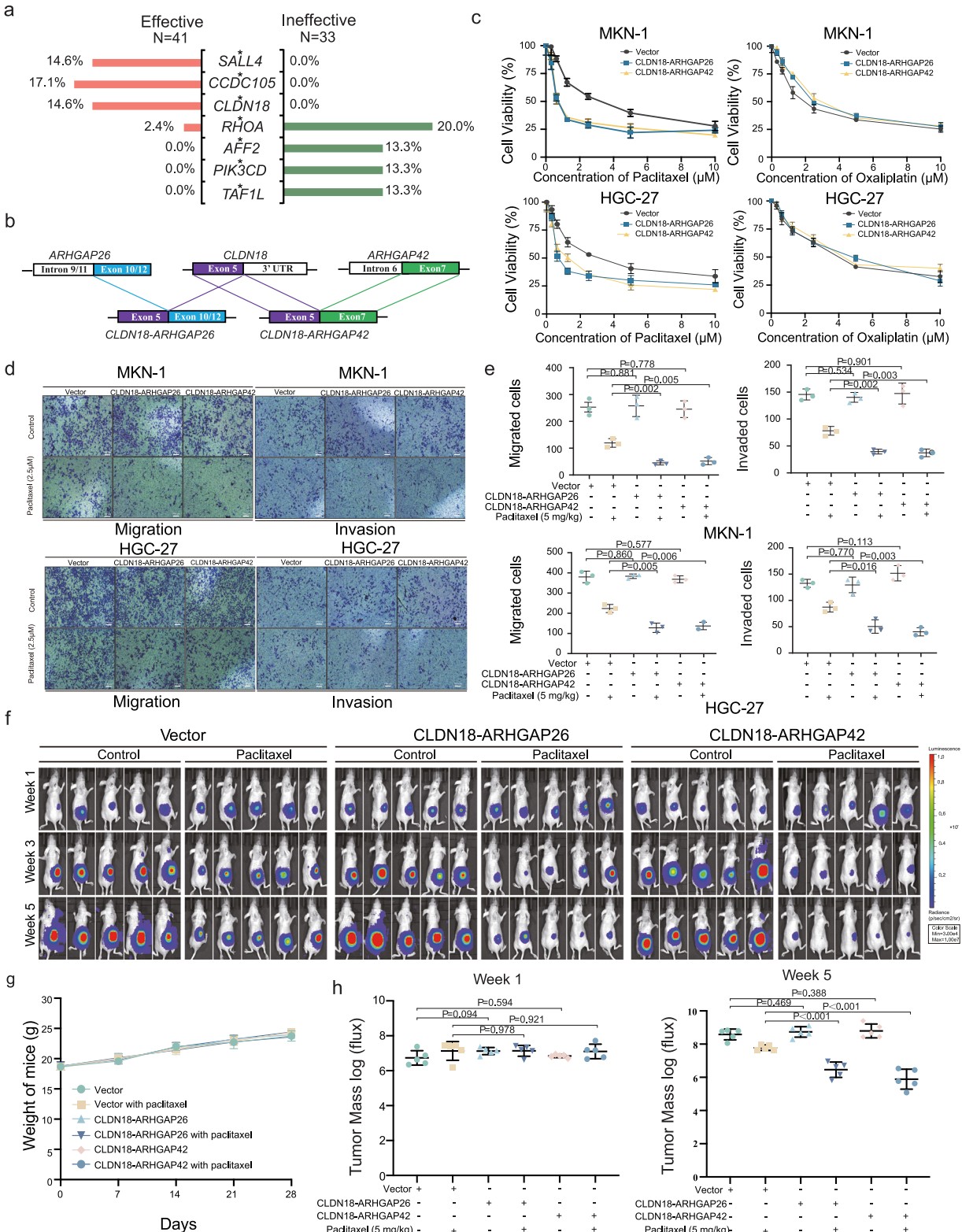

Synchronous metastasis and metachronous metastasis are two different modes of tumor metastasis. Kim et al. found that the mutational profiles of synchronous and metachronous metastases of colorectal cancer were similar, corresponding to their similar clinical outcomes[25]. However, the clinical outcomes of synchronous and metachronous ovarian metastases from GC were significantly different[26]. In our cohort, comparative genomic analysis between cases of synchronous and metachronous metastasis revealed a limited number of genes with significantly distinct mutation profiles. The disparities were primarily observed in the mutation frequencies of the *ERBB3* and *PIK3CA* genes, alongside copy number variations (CNVs) on chromosomes 5, 7, and 17. These differences may contribute to the treatment disparities observed between these two patient groups. The high proportion of linear evolution in synchronous metastasis group and the high proportion of intermediate evolution in the metachronous metastasis group maybe correspond to the different

**Fig. 6 | CLDN18 fusion mutations were associated with paclitaxel efficacy. a** A correlation analysis between mutated genes and paclitaxel efficacy. The percentages represent the proportion of patients harboring mutations within each respective group. Chi-squared test ($\chi^2$) and Fisher's exact test were used in the comparison. **b** A schematic diagram of *CLDN18* fusion in metastatic ovarian lesion. **c** The results of CCK-8 assays following treatment with paclitaxel (0, 0.3125, 0.625, 1.25, 2.50, 5.0, 10.0 uM) or oxaliplatin (0, 0.3125, 0.625, 1.25, 2.50, 5.0, 10.0 uM) for 48 h in MNK-1 and HGC-27 GC cell lines with CLDN18-ARHGAP26/42 fusion mutations (n = 3 biological replicates). Source data are provided as a Source Data file. **d** The results of transwell following treatment with paclitaxel (2.5 uM) or oxaliplatin (2.5 uM) for 72 h in MNK-1 and HGC-27 GC cell lines with CLDN18-ARHGAP26/42 fusion mutations (n = 3 biological replicates). **e** Quantitation of the transwell (n = 3 biological replicates). *P*, two-sided Student's *t*-test. Source data are provided as a Source Data file. **f** MKN-1 GC cells stably transfected with CLDN18-ARHGAP26/42

fusion mutations or empty vector were subcutaneously inoculated into the left ovary nude mice (n = 5 biological replicates). One week later, mice were treated with 10 mg/kg/tiw paclitaxel for 4 weeks. The luciferase signals in the mice were detected and images were obtained using an IVIS imaging system. *P*, two-sided Student's *t*-test. **g** The mice were monitored for changes in body weight as a surrogate marker for toxicity. There is no significant difference between any two groups at the same time point. *P*, two-sided Student's *t*-test. Source data are provided as a Source Data file. **h** The average tumor mass (determined by the detected photons/sec) of mice in different groups at week 1 (beginning of intervention) and week 5 (end of intervention, n = 5 biological replicates). *P*, two-sided Student's *t*-test. In **c**, **e**, **g** and **h**, error bars represent mean ± standard deviations. In **e**, **h**, the boxplot elements indicate the maxima, 75th percentile, median, 25th percentile, and minima. Source data are provided as a Source Data file.

differentiation time of metastatic events. In the linear evolution model, metastatic cells with metastatic clonal ability appear in the late stage of tumorigenesis, so there is a higher mutation consistency between the primary and metastatic lesions[18]. In the parallel evolution model, metastatic lesion appears at the early stage of the development of the primary tumor. The primary tumor and metastatic tumor clones continue to evolve in parallel under different pressures, resulting in obvious genetic differences between the primary and metastatic lesions[18,27]. Interestingly, we found that patients in parallel evolution group were more sensitive to paclitaxel chemotherapy and had a better prognosis. Evolutionary classification can provide prognostic and therapeutic guidance for GC patients with ovarian metastasis. Furthermore, we found more enrichment of signaling pathway mutations and more primary gastric specific mutations in patients with synchronous metastasis. These results revealed the different molecular characteristics between synchronous and metachronous ovarian metastasis, which supported different treatment strategies for different metastatic subtypes.

Chemotherapy stands as the primary therapeutic approach for GC patients with ovarian metastasis, and among the available chemotherapy regimens, 5-Fu, oxaliplatin, and paclitaxel typically emerge as the foremost choices for the GC patients[28–30]. In our previous study, we observed that paclitaxel-based chemotherapy improved the survival of GC patients with peritoneal or ovarian metastases[19]. Although systemic chemotherapy can provide symptom palliation and prolonged survival in patients with ovarian metastasis, the efficacy was disappointing with a median survival time of 8 to 14 months[3]. In this study, we have uncovered potential biomarkers that can be utilized for predicting the effectiveness of paclitaxel treatment. Notably, mutations in *CCDC105*, *SALL4*, and *CLDN18* were found to be associated with positive chemotherapy responses, while mutations in *AFF2*, *PIK3CD*, *RHOA*, and *TAF1L* were linked to a lack of chemotherapy response. As a pivotal regulator in cell cycle, downregulation of SALL4 has been observed to enhance sensitivity to chemotherapy in breast and lung cancers[31,32]. *PIK3CA* mutation as one of the major driver oncogenes in cancer, the significance of *PIK3CA* mutations in cancer has been elucidated in many studies[33]. Wang et al. confirmed that colorectal cancer patients with *PIK3CA* mutation showed worse response to first-line chemotherapy than those without *PIK3CA* mutation, which may be related to the activation of the PI3K-Akt signaling pathway caused by mutations in exon 9 and 20 in *PIK3CA*[33]. *RHOA* mutations are associated with the development of diffuse-type GC and participate in various cellular process, including the regulation of the cellular cytoskeleton and the contractile ability of actin-myosin[34,35]. Changhwan et al. confirmed that RHOA pathway inhibition can reverse the 5-FU and cisplatin chemotherapy resistance[36]. However, the function of these features and clinical significance in the treatment of ovarian metastasis from GC still needs to be further verified.

CLDN18 as a potential target for GC treatment, it has been identified to correlate with tumor size, aggressiveness, potential metastasis,

and prognosis in GC patients[22,23,37]. Currently, numerous clinical drugs targeting CLDN18 are undergoing clinical research. The SPOTLIGHT clinical trial has confirmed that targeting CLDN18.2 with zolbetuximab significantly extended progression-free survival and overall survival when administered in combination with mFOLFOX6 compared to placebo plus mFOLFOX6 in patients diagnosed with CLDN18.2-positive, HER2-negative, locally advanced unresectable, or metastatic gastric or gastro-esophageal junction adenocarcinoma[38]. In recent years, recurrent structure rearrangement has been identified frequently between CLDN18 and ARHGAPs[5,39]. The prevalence of CLDN18-ARHGAP fusion mutations in gastric cancer has been reported to be about 20%[20]. The CLDN18-ARHGAP fusion-positive patients typically exhibit larger tumors and a higher incidence of metastatic lymph nodes, and therefore often diagnosed at more advanced stage than patients without CLDN18-ARHGAP fusion[40]. Shu et al. confirmed that patients with CLDN18-RHGAP fusion had poor survival benefit and did not derive significant benefits from oxaliplatin/fluorouracil chemotherapy[20]. Interestingly, our study also found that GC patients bearing ovarian metastasis with CLDN18-ARHGAP fusion demonstrated resistance to oxaliplatin-based chemotherapy while displaying sensitivity to paclitaxel-based treatment. However, the specific mechanisms through which CLDN18-ARHGAP mutations influence chemotherapy sensitivity remain unclear at present. The Rho-GTPase activity of ARHGAP facilitates the conversion of active RHOA into its inactive form[41]. The CLDN18-ARHGAP fusion can reduce cell-ECM adhesion via inhibiting the RHOA activity through recruiting GAP activity of ARHGAP into the vicinity of the plasma membrane of CLDN18. The presence of the CLDN18-ARHGAP26 mutation has been shown to facilitate the progression and metastasis of gastric cancer (GC) through the loss of CLDN18 function and the acquisition of ARHGAP26 functions[42]. In contrast, we found that CLDN18-ARHGAP fusion had no significant effect on the invasion and migration ability of GC cells. We also found that CLDN18-ARHGAP fusion mutations was not associated with CLDN18 expression. It will be intriguing to investigate how CLDN18-ARHGAP fusion impacts cellular proliferation and the effectiveness of chemotherapy in GC patients with ovarian metastasis.

There are some limitations in this study. Firstly, our study was a single-center study with a relatively small sample size due to the scarcity of the disease. Although this study represented large genomic analyses of primary gastric and matched ovarian metastasis samples, our analyses were still limited by sample size. Chromosomal and genetic alterations need to be fully validated in a larger multi-center cohort. Secondly, because of the retrospective study design, the sample quality could not meet the requirement of multi-omics analysis, which might provide profiling of molecular characteristics in ovarian metastasis from GC. Finally, although we have validated the correlation between *CLDN18* fusion and paclitaxel sensitivity in vitro and *vivo* models, further research is needed on the biological mechanisms by which *CLDN18* fusion promotes paclitaxel sensitivity, and therefore personalized treatment strategies need to be developed.

In summary, our research marks the elucidation of distinct mutational characteristics existing between primary gastric tumors and metastatic ovarian tumors, as well as between synchronous and metachronous ovarian metastases. Notably, we have uncovered a correlation between the CLDN18-ARHGAP fusion mutation and efficacy of paclitaxel chemotherapy. These findings contribute to a more profound understanding of the mutational attributes intrinsic to this tumor entity, offering a molecular foundation for precision therapeutic strategies.

## Methods

### Ethics approval

This research complies with all relevant ethical regulation. The study of human tumor samples was performed according to the declaration of Helsinki and Good Clinical Practice and approved by the Ethical Committee of Zhejiang Cancer Hospital (IRB-2022-279). Informed written consent was obtained from all participants. The animal experiments were conducted with the approval of the Animal Ethical Committee at the institute of Zhejiang Chinese Medical University (202110-0682).

### Patient enrollment

From January 1, 2012 to December 31, 2020, 192 patients with ovarian metastasis were screened from 15,315 GC patients. After excluding 101 patients who did not receive ovarian resection or biopsy and 17 patients with insufficient tumor volume, 74 patients were finally enrolled for this study (Fig. 1a). A total of 65 primary gastric and 73 ovarian metastatic tumor samples were collected from the enrolled patients, including 64 pairs matched primary gastric and metastatic ovarian tumor samples. Follow-up data were obtained by phone and through an out-patient clinical database. The last follow-up occurred in January 2022, and follow-up data were available in 71 (71/74, 95.9%) patients. The overall survival (OS) time was calculated from the date of diagnosis to the last day of follow-up or the date of death. Patients who received paclitaxel-based chemotherapy were evaluated for therapeutic efficacy. The response to treatment was evaluated according to the criteria of Response Evaluation Criteria in Solid Tumors (RECIST 1.1). Patients with complete response (CR) or partial response (PR) were considered as effective, and patients with stable disease (SD) or progressive disease (PD) were considered as ineffective.

### Sample Collection, WES detection, and GA identification

Tumor tissue samples and matched blood samples were collected for the determination of genomic alterations (GAs). Genomic DNA was isolated using a QIAamp DNA FFPE Tissue Kit and a QIAamp DNA Blood Midi Kit (Qiagen, Hilden, Germany), according to the manufacturer's instructions. The concentration of DNA was measured using Qubit and normalized to 20–50 ng/μL. WES libraries were prepared and captured using the SureSelect Human All Exon V6 kit (Agilent Technologies), according to the manufacturer's instructions, and then sequenced using an Illumina HiSeq X Ten system (Illumina, Inc., CA). WES was conducted with a mean coverage depth of 187x (range: 108-344x) for tumor samples, consistent with recommendations.

SNVs were identified using MuTect (v1.17). Insertions/deletions (Indels) were identified using PINDEL (V0.2.4). The functional impact of these mutations was annotated using SnpEff3.0. CNVs were identified using Control-FREEC (v9.4), with the following parameters: window = 50,000 and step = 10,000. Gene fusions were detected using an in-house pipeline[43]. Gene rearrangements were assessed with the Integrative Genomics Viewer.

### Mutational signature analysis

According to the number of different types of point mutations such as C > A/G > T, C > G/G > C, C > T/G > A, T > A/A > T, T > C/A > G, and T > G/A > C, a cluster analysis was performed in order to observe similarity in tumor samples. Extracted mutational features were compared with the pan-cancer catalog for 94 known features cited in the cancer somatic mutation catalog (COSMIC) database (https://cancer.sanger.ac.uk/signatures/) using Mutational Patterns packages (3.6.0)[44]. The similarity of mutational features was assessed based on a cosine similarity > 0.85, which indicated common features.

### Integrated pathway analysis

For each designated group−primary, metastasis, synchronous, metachronous, and TCGA cohorts−we ascertained the top 100 genes that demonstrated the highest mutation frequencies. These gene cohorts were then subjected to pathway enrichment analysis utilizing the Kyoto Encyclopedia of Genes and Genomes (KEGG) database to elucidate the significant biological pathways implicated in each group. Gene set clustering into corresponding KEGG pathways was executed via the DAVID (Database for Annotation, Visualization and Integrated Discovery) Bioinformatics Resources 6.8 platform. Pathways with FDR < 0.05 were defined as significantly enriched in GC.

Following initial pathway categorization, we aggregated the enriched pathways from all groups to refine our focus on biological functions. Comparative pathway significance was evaluated by quantifying the prevalence of relevant genes, and a pathway score was computed for each group using the formula: Pathway Score = (Summation of mutations across genes within the pathway)/(Total gene count within the pathway × Patient cohort size). Then the signaling pathways were clustered and ranked in descending order according to their respective scores.

After excluding some signaling pathways with high redundancy in mutations, and focusing on these related to cell proliferation, migration and differentiation, a chart combining gene frequency and bidirectional regulatory relationships of each gene was constructed to analyze the complex interactions and regulatory mechanisms between key genes and their related upstream and downstream mediators in these pathways.

### Phylogenetic Tree

All SNVs were used to construct phylogenetic tree based on the Lineage Inference for Cancer Heterogeneity and Evolution (LICHeE) method[45]. Relying on the phylogeny model[46], LICHeE utilized the somatic SNV patterns of samples and their VAFs as lineage markers for reconstructing a phylogenetic tree. The genetic distance between all pairs of samples in each patient were calculated using Treeomics. Treeomics v1.9.0[47] were used with the default settings to reconstruct the phylogenies of the metastatic tumor using high-quality somatic variants and the CN alterations were identified by WES, respectively.

### Immunohistochemistry

The expression of CLDN18 in the ovarian metastasis and paired primary gastric tumor was performed via IHC. The IHC staining with antibodies against CLDN18 (#ab222512, Abcam, dilution ratio 1:200). The tissue sections were dewaxed and rinsed with distilled water, followed by antigen retrieval. Subsequently, the tissue sections were rinsed three times for 5 min each with PBS. The tissue microarrays were incubated overnight at 4 °C with antibodies against CLDN18 (#ab222512, Abcam, dilution ratio 1:200), followed by washing with PBS three times for 5 min each. Then, the appropriate secondary antibodies, i.e., goat anti-rabbit IgG H&L (PV-9003, ZSGB-BIO Corp., dilution ratio 1:1000) were added. After incubation for 30 min, the slides were washed with PBS for 5 min three times. DAB color development and haematoxylin staining of cell nuclei were performed using a DAB color development kit (ZLI-9065, ZSGB-BIO Corp., Shanghai, China). Two experienced pathologists who were blinded to the clinical outcomes performed the scoring independently. CLDN18 positive was defined as moderate to strong expression in ≥40% of tumor cells.

## Cell lines

Human GC cell lines MKN-1 and HGC-27 (obtained from Shanghai Bioleaf Biotech Co., Ltd., Shanghai, China), were derived from female GC patients with undifferentiated adenocarcinoma. These cell lines exhibited robust metastatic and invasive capabilities and demonstrated insensitivity to chemotherapy. Authentication of these cell lines was conducted using Short Tandem Repeat analysis, and regular testing was carried out to ensure the absence of mycoplasma contamination.

## Gene overexpression by lentiviral vector transduction

Lentivirus containing CLDN18-ARHGAP26/CLDN18-ARHGAP42 overexpression constructs or a negative control was synthesized by Gene-Chem Biotechnology Company (Shanghai, China). Lentiviral transduction of GC cells was performed following the manufacturer's instructions. After 72 h stable cell lines were selected using 1 μg/mL puromycin. Transfection efficiency was evaluated by western blot analysis.

## Cell viability assay

CCK-8 (GLPBIO, United States) assays were conducted to measure cellular viability. The transfected MKN-1 and HGC-27 cell lines were seeded into 96-well culture plates. After 12 h, cells were exposed to the paclitaxel (0, 0.3125, 0.625, 1.25, 2.50, 5.0, 10.0 uM) or oxaliplatin (0, 0.3125, 0.625, 1.25, 2.50, 5.0, 10.0 uM) for 48 h and incubated with CCK-8 reagent for 3 h. Thereafter, a microplate reader (Thermo Varioskan LUX, MA, United States) was used to measure the absorbance (OD) at 450 nm.

## Transwell migration and invasion (Matrigel) experiments

For migration assays, $1 \times 10^5$ cells in 200 μL of serum-free media containing paclitaxel (2.5 uM) were seeded in the upper chamber of an insert (8 μm pore size, Corning, USA). For invasion assays, $1 \times 10^5$ cells in 200 μL of serum-free media containing paclitaxel (2.5 uM) were seeded in the upper chamber of an insert coated with Matrigel (BD Biosciences, San Diego, CA). Then, 600 μL of medium containing 20% FBS was added to the lower chamber. After incubation for 72 h, the cells attached onto the upper side of the transwell were mechanically removed with a cotton stick. Next, the cells on the bottom surface of the membrane were fixed with 4% paraformaldehyde for 10 min and then stained with 0.4% crystal violet solution for 10 min. Images of the migrated and invaded cells were captured with a Nikon Digital Sight DS-L1 camera.

## Ovarian metastasis models

To establish the ovarian metastasis models, 4-week-old female nude mice were injected with $3 \times 10^6$ GC cells transfected with the luciferase lentiviral vector (MKN-1-luc) (OV-NC, OV-CLDN18-ARHGAP26, or OV-CLDN18-ARHGAP42) mixed with 10 μL PBS and 10 μL Matrigel (BD Biosciences) into the left ovary. One week later, mice were received intraperitoneal injections of paclitaxel at a dose of 10 mg/kg/tiw for 4 weeks. Throughout the experiment, the body weight of the mice and the fluorescence intensity of the tumors were monitored. In vivo, imaging was performed twice a week using a Xenogen IVIS 200 imaging system (Caliper Life Sciences, USA) after intraperitoneal administration of fluorescein substrate (150 mg/kg). The tumor inhibition rate was determined using the LT Living Image 4.3 Software.

## Statistical analysis

R version 3.6.1 (R Foundation for Statistical Computing, Vienna, Austria) and SPSS version 24.0 (IBM Corp., Armonk, NY, USA) was used for statistical analyses. Chi-squared test ($\chi^2$) and Fisher's exact test were used in the comparison of the gene alteration frequency between primary and metastasis or synchronous and metachronous groups. Fisher's exact tests were also used to analyze the significance of response between different evolution groups. The differences in the distribution of mutations between synchronous metastasis and metachronous metastasis were compared using a one-way ANOVA analysis of variance. Student's t-test was used to compare the invasion and migration ability of different GC cells. Overall survival was estimated using the Kaplan-Meier method, and differences between groups were assessed by the log-rank test. A $p$ value < 0.05 was considered statistically significant.

## Reporting summary

Further information on research design is available in the Nature Portfolio Reporting Summary linked to this article.

## Data availability

The WES data were deposited in the Sequence Read Archive (SRA) database under the accession number PRJNA1095670, which is publicly available. The variant data generated in this study have been deposited in the China National Center for Bioinformation (CNCB) database under accession code HRA007045 [https://ngdc.cncb.ac.cn/gsa-human/s/bgJlk98A], which is publicly available. Public datasets used in this study include the KEGG databases [https://www.kegg.jp/kegg/kegg1.html] and TCGA (Stomach Adenocarcinoma, Firehose Legacy) [https://www.cbioportal.org/study/summary?id=stad_tcga] studies were downloaded from cBioPortal. All data necessary to evaluate the conclusions in the paper are provided in the paper and/or the Supplementary Information. Source data are provided as a Source Data file. Source data are provided with this paper.

## Code availability

This study did not generate any unique code. All software and algorithms used in this study are freely or commercially available and are listed in the methods section.

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

## Acknowledgements

We acknowledge the support from Haiyan Wu and Baofeng Lian for their suggestions on analytical methods. We are grateful to all the study participants, patients, and their family members for their contributions and support. This work was supported by the Natural Science Foundation of Zhejiang Province of China (No. LBZ22H160002 to P.F.Y.)

## Author contributions

All authors have full access to all data used in the study and take responsibility for the integrity of the data and the accuracy of the data analysis. P.F.Y., C.H., G.Y.D., X.L.S., Y.C., X.L.C., W.W. and Q.X. acquired experimental data. J.Q.F., X.M.H., S.H.Y., H.C., Z.Z.W., L.H., F.P. and Y.A.D. provided administrative, technical or material support. P.F.Y. and X.D.C. were involved in the study design and obtaining funding.

## Competing interests

The authors declare no competing interests.
