## [Peer Review File · Nature Communications]

Mutation characteristics and molecular evolution of ovarian metastasis from gastric cancer and potential biomarkers for paclitaxel treatmentREVIEWER COMMENTS

Reviewer #1 (Remarks to the Author): expertise in WES of gastric cancer

This manuscript present data on a relatively large set of matched primary gastric cancers and ovarian metastases. It demonstrates considerable genetic heterogeneity between the primary tumors and the metastases, and notes some differences between synchronous and metachronous metastases. The statistical analyses are often however not convincing. Also, unfortunately, the description of the evolutionary models is confusing and their depiction (Fig 4) is not clear. These observations on the mutation frequency differences between the primary and metastatic lesions are of great interest, but need to be described and depicted more clearly. The sequencing methods are sound, and well described.

Major comments

1. The abstract states that mutations in FUS, ETV4, CBF4 etc are frequent in ovarian lesions, but line 178 in the results states the opposite.
2. Lines 165-: there is no formal statistical analysis to show the stated pathways are 'enriched' for mutations.
3. Line 191: what does 'screened out' mean? How were these signatures identified? Fig 2H marks 10 patients as being over 50yrs. If the median age is 46yrs, surely there are more patients over 50?
4. Lines 199-200: The statement that the enrichment of pathway mutations in primary GCs is generally consistent with TCGA data is not supported by the data provided in the boxes in Fig, 3 (eg. MYC: 3.1% vs 13.2%; KRAS: 0% vs 8.2% for the activated mutations in primary GCs vs TCGA, respectively).
5. Line 211-: The most common shared mutations are not well described. What would be good to see is the proportion of mutations in each frequently mutated gene that are present in both the primary tumor and the metastasis. Ie what mutations are absolutely required? Showing the common mutations with the number of tumors (64) as the denominator really just tells us about overall mutation frequency. Line 286 tells us that RhoA is always present in the primary tumour and the metastasis. This is very interesting, but needs to be presented properly alongside the other commonly mutated genes.
6. It was difficult to reconcile the test (line 211-230) with Figure 4. It also wasn't clear how the three groups were defined. Were arbitrary cutoffs used, or did the data naturally fall into three groups? The figure legend for Fig 4 also did not explain the figure well.
7. Line 236: the proportion of what?
8. Line 239: This result isn't significant ($p=0.067$).
9. Lines 248-249: PIK3CA and TMEM132D aren't significant as stated.
10. 10. Lines 255, 258: What test has been done to say mutations are 'enriched'?
11. Line 274-: Multiple testing corrections don't appear to have been used when comparing the mutation frequencies between groups. These genes may not be significant after that.
12. The Lauren classification should be provided in Table 1 if possible.

Minor comments

Line 160: GAs not Gas

Line 183: Figure 2E not 1E

Line 286: the meaning of "partially arising" evolution isn't clear

Table 2: NTRK1 and RAD51 aren't present in Fig 2A.

Reviewer #2 (Remarks to the Author): WES cancer evolution expertise

Please see the attached document for my comments.

The paper by Yu et al. characterized the molecular landscape of ovarian metastasis from gastric cancer and identified the significantly mutated genes. They further elucidated the molecular heterogeneity between synchronous metastasis and metachronous metastasis. Finally, they identified potential biomarkers for predicting the efficacy of paclitaxel treatment and found CLDN18 fusion to be a potential predictive biomarker for paclitaxel treatment response irrespective of CLDN18 expression. Overall, the paper is neatly written and problem statement is novel. However, I have some concerns addressing which is necessary.

1. The authors mentioned that the enrichment of pathway mutations of primary GC lesions in their cohort is consistent with that of TCGA cohort. However, the DAVID/KEGG pathway analyses have not been properly presented (e.g., see Fig. 5C in <https://www.nature.com/articles/nature13480>). Can the authors present the heatmap for pathway analyses for their cohort as well as TCGA? The authors may have chosen main signaling pathways known in literature and compared them across the cohorts. However, it will be important to perform an unbiased pathway analyses to see what pathways were enriched in their cohort based on the mutations identified. Also, for Figure 3, the source for the pathways is not mentioned. The authors need to provide more details in the method section how they obtained the pathway for Fig. 3.
2. Based on the phylogenetic analysis, the authors classified the patients into 4 categories. However, this classification seems to be just based on the number of mutations on the trunk or the branches. Can this classification be improved by fitting a mathematical tumor evolution model for each patient? For figure 4, the caption needs to be elaborated. What are the Venn diagrams representing? Are these trees corresponding to some example patients? Was there any difference in survival for these four classes?
3. In Figure 6, the authors compared the frequency of mutations in genes involved in certain pathways. How these pathways were selected is not clear from the text. Can the authors perform unbiased pathway analysis to show how same/different pathways are enriched for synchronous vs metachronous metastasis?
4. In line no 283-289, the authors classified the evolution of genes associated with ineffective Paclitaxel therapy. However, the basis of this classification is not elaborated. The authors need to elaborate hoe this classification was performed and the results need to be visualized.
5. Line no 304-312, the authors mentioned that the expression of CLDN18 was positive in some samples and negative in some other samples. It is not clear how expression can be negative. These statements need to be properly explained. The authors claim CLDN18 fusion to be a predictive biomarker for paclitaxel treatment response, however, this claim is not supported by data. While the authors show that there is no relationship between CLDN18 fusion and expression, how CLDN18 fusion is helpful in predicting the effectiveness of paclitaxel therapy is not clear. Can the authors perform any experiment to show this association?
6. The fusion detection method is not described in detail. Since this is an important component leading one of the main findings, the authors need to elaborate the details.
7. The paper contains grammatical errors which need to be fixed.

Reviewer #3 (Remarks to the Author): clinical expertise in gastric cancer ovarian metastasis

This study provides deeper insight into mutation characteristics of this tumor entity, which could be a molecular basis for precise treatment. Here are some shortcomings I suggest revising:

1. This study lacks some wet experiments. I hope the author can add some experiments. It seems that the workload of this study is slightly small and there are some key experiments lacking.
2. Can some experiments be added to illustrate the grouping of patients with partial or complete remission mentioned by the author in the method, such as immunohistochemistry?
3. Line No.183, the author mentioned Fig.1E, but the submitted figures does not contain Fig.1E. I hope the author can check your manuscript carefully.
4. Line No.188, the author mentioned Fig.2F, however, Figure 2 does not have an F diagram.
5. Line No.301~302 and No. 304~312, the number of samples is too small and the experimental data is insufficient to serve as strong evidence. Please improve the experiment.
6. I hope the author will add more information about ovarian metastasis of gastric cancer in the introduction.
7. Please explain how to determine whether the sample is gastric cancer with ovarian metastasis or primary ovarian cancer.
8. Line No.627, there are two descriptions of F in the description of Fig.2, but there is no F in Figure 2. Please check the manuscript again.
9. Can there be further in-depth validation of targeted genes? Such as adding some cellular functional experiments, etc.

Recommendation: Rejected

RESPONSE TO REVIEWERS' COMMENTS

Reviewer #1:

Comment 1: The abstract states that mutations in FUS, ETV4, CEBF, etc are frequent in ovarian lesions, but line 178 in the results states the opposite.

Reply: Thank you for bringing this discrepancy to our attention. Due to the word limit of the abstract, we have deleted this sentence and retained the description in the article, and changed the abstract to the following: “Ovarian metastasis is one of the major causes of treatment failure in patients with gastric cancer (GC). However, little is known about the genomic characteristics of ovarian metastasis of GC. In this study, we enrolled 74 GC patients with ovarian metastasis, of which 64 had matched primary and metastatic samples. We comprehensively characterized the mutation landscape of this disease and investigated the molecular heterogeneity and pathway mutation enrichments between synchronous and metachronous metastasis. Furthermore, we classified patients into distinct clonal evolution patterns based on the distribution of mutations in paired samples. Notably, the parallel evolution group exhibited the most favorable prognosis. Additionally, by analyzing the differential response to chemotherapy, we identified potential biomarkers, including *SALL4*, *CCDC105*, and *CLDN18*, etc., for predicting the efficacy of paclitaxel treatment. Furthermore, we experimentally validated that *CLDN18* fusion mutations can improve tumor response to paclitaxel treatment in GC with ovarian metastasis *in vitro* and *in vivo*.” (Line 30-43, Page 1)

Comment 2: Lines 165-: there is no formal statistical analysis to show the stated pathways are ‘enriched’ for mutations.

Reply: Thank you for your comment and feedback. To address this concern, we conducted a comprehensive statistical analysis to determine the enrichment of

mutations in all pathways. We have made the revision as follows: “After classifying all primary gastric and metastatic ovarian samples by metastasis subtype, we further integrated the CNV and mutation data of these four groups of samples to characterize genomic changes in enriched signaling pathways and compared them with GC samples in the TCGA database (Fig 4a, Supplementary Table 2).” (Line 171-181, Page 8)

Comment 3: Line 191: what does ‘screened out’ mean? How were these signatures identified? Fig 2H marks 10 patients as being over 50yrs. If the median age is 46yrs, surely there are more patients over 50?

Reply: Thank you for your questions and comments.

① For the first question, “screened out” means that the similarity of mutational features was evaluated using a cosine similarity threshold of > 0.85 , indicating the presence of common features.

② For the second question about how the signatures were identified, we compared the extracted mutational features with the pan-cancer catalogue, which consists of 94 known features cited in the cancer somatic mutation catalogue (COSMIC) database (<https://cancer.sanger.ac.uk/signatures/>). This comparison was performed using the Mutational Patterns packages (version 3.6.0)¹.

③ For the third question, we carefully reviewed our data and ensured that the correct number of patients over 50 years old was accurately represented in the revised Fig. 2g (Fig. 2H in the original manuscript).

Comment 4: Lines 199-200: The statement that the enrichment of pathway mutations in primary GCs is generally consistent with TCGA data is not supported by the data provided in the boxes in Fig. 3 (eg. MYC: 3.1% vs 13.2%; KRAS: 0% vs 8.2% for the activated mutations in primary GCs vs TCGA, respectively).

Reply: Thank you for your questions and comments. We deleted this sentence in the revised manuscript and further examined the differences between each gene in the two groups of patients. We have made the revision as follows: “In primary gastric lesions, the most frequently altered genes included *TP53* (46.2%), *ERBB2* (27.7%), *CDHI* (20.0%), *ERBB3* (13.8%), and *PIK3CA* (13.8%), and were enriched in the RAP1, PI3K-AKT, MAPK and RAS pathways (Fig. 4b). For metastatic ovarian lesions, the most frequently altered genes included *TP53* (45.2%), *PIK3CA* (20.5%), *CDHI* (19.2%), *TGFBR2* (16.4%), and *ERBB2* (13.7%), and were enriched in the FOXO, P53 and ERBB pathways (Fig. 4b). The most frequently altered genes included *TP53* (44.5%), *PIK3CA* (20.5%), *ERBB2* (18.1%), *CDKN2A* (16.7%), *KRAS* (15.8%) and *APC* (14.0%)

and were enriched in the ERBB, PI3K-AKT and Wnt pathways in the TCGA GC database (Fig. 4b).” (Line 171-180, Page 8)

Comment 5: Line 211-: The most common shared mutations are not well described. What would be good to see is the proportion of mutations in each frequently mutated gene that are present in both the primary tumor and the metastasis. Ie what mutations are absolutely required? Showing the common mutations with the number of tumors (64) as the denominator really just tells us about overall mutation frequency. Line 286 tells us that RhoA is always present in the primary tumour and the metastasis. This is very interesting, but needs to be presented properly alongside the other commonly mutated genes.

Reply: Thank you for your insightful comment and suggestion.

① For the first question, to address this concern, we provided a more comprehensive analysis of the shared mutations in the following figure, including the proportion of mutations in each frequently mutated gene that are present in both the primary tumor and the metastasis. This will allow for a better understanding of the mutations that are consistently present and potentially required for tumor progression and metastasis. We have made the revision as follows: “The most frequently occurring mutations observed in shared, primary and metastatic tumors were *TP53* (22/30/33), *ARID1A* (14/22/29), *CDH1* (11/13/14), *TTN* (8/10/21), *ERBB2* (7/18/10), and *TGFBR2* (7/7/12).” (Line 190-193, Page 9)

② Furthermore, we will present the information about *RHOA* being always present in both the primary tumor and the metastasis alongside the other commonly mutated genes. This will provide a more complete and comparative analysis of the frequently mutated genes and their presence in both tumor sites. We have made the revision as follows: “In addition, we also investigated the mutation of these genes in paired primary gastric and metastatic ovarian lesions, and found that the

evolution of these genes could be classified into the following three subtypes (Supplementary Fig. 7): 1) an subtype completely arising from primary gastric lesions, such as *RHOA* mutation (6 mutations arising from primary gastric lesions/6 mutations in metastasis tumors) and *TAF1L* mutations (1/1); 2) an subtype partially arising from primary gastric lesions, such as *SALL4* (2/3), *CCDC105* (1/3), *CLDN18* (4/7), and *PIK3CD* (1/3); and 3) an subtype newly developed in metastatic lesions, such as *AFF2* mutation (0/1).” (Line 251-258, Page 12)

Comment 6: It was difficult to reconcile the test (line 211-230) with Figure 4. It also wasn't clear how the three groups were defined. Were arbitrary cutoffs used, or did the

data naturally fall into three groups? The figure legend for Fig 4 also did not explain the figure well.

Reply: Thank you for your questions regarding the test and Figure 4 in the original manuscript.

- ① For the first question, we have redrawn Figure 4 from the original manuscript into Figure 5 in the revised manuscript, where a new classification method² is used to divide all patients into three groups: linear evolution, intermediate evolution, and parallel evolution.
- ② For the second question, in the new classification system, the three groups are distinguished using genetic distance, where genetic distance > 0.7 is defined as linear, genetic distance less than 0.4 is defined as parallel, and genetic distance between 0.4 and 0.7 is defined as intermediate (Fig. 5a).
- ③ For the third question, we have redrawn Figure 4 from the original manuscript into Figure 5 in the revised manuscript, we have made the revision as follows: “Among the 55 patients with a well-defined evolutionary relationship were classified into three distinct migration patterns, including parallel evolution, linear evolution and intermediate evolution (Fig. 5a). There was a few shared GAs in the paired samples of 18 patients, with a median genomic distance of 8.0% and a range of 0.8% to 29.3%, which was defined as parallel evolution. The parallel evolution shared a short trunk, and the phylogenetic tree involved multiple branches from a founder clone, suggesting that tumor cells from metastatic and primary tumors of these patients developed independently at an early stage. The paired samples of 20

patients had more shared mutations, with a median genomic distance of 62.7% and a range of 25.6% to 84.8%, which was defined as linear evolution. The linear evolution shared a long trunk that harbored the initial cancer driver alteration, indicating that a founder clone that acquired driver alteration disseminated late from the primary tumor and evolved into the metastatic tumor. Seventeen patients with a median number of mutations between the trunk and branches were considered as intermediate evolution (Fig. 5a).” (Line 197-210, Page 9)

Comment 7: Line 236: the proportion of what?

Reply: Thank you for your questions and comments. We have changed the wording in the revised manuscript to more clearly describe the phenomena we have observed. We have made the revision as follows: “The proportion of shared mutations between primary gastric and metastatic ovarian tumors was 24.9%, ranging from 0.0% to 84.8% (Supplementary Table 3).” (Line 188-189, Page 9)

Comment 8: Line 239: This result isn't significant ($p=0.067$).

Reply: Thank you for your comment. We have changed the wording in the revised manuscript to more accurately describe the differences we have observed. We have made the revision as follows: “We found that patients with synchronous metastases exhibited a relatively lower proportion of specific mutations in the primary GC ($P = 0.046$). However, there was no significant difference in the proportion of specific mutations in metastatic ovarian lesion and shared mutations between patients with synchronous and metachronous metastases (Fig. 3d).” (Line 161-165, Page 7)

Comment 9: Lines 248-249: *PIK3CA* and *TMEM132D* aren't significant as stated.

Reply: Thank you for your comment. We have removed the direct comparison between synchronous and metachronous patients and added a comparison of primary or metastatic lesion variants between synchronous and metachronous patients. We have made the revision as follows: “Compared with synchronous patients in the primary lesion sample, the mutation frequency of *PIK3CA* was significantly higher in metachronous patients (Fig. 3b). In metastatic lesions, the frequency of *ERBB3* mutations is significantly higher in synchronous patients compared to metachronous patients (Fig. 3b)” (Line 151-154, Page 7)

Comment 10: Lines 255, 258: What test has been done to say mutations are ‘enriched’?

Reply: Thank you for your comment and feedback. We conducted a comprehensive statistical analysis to determine the enrichment of mutations in all pathways. We included the appropriate statistical tests and provided the necessary results and interpretation in the revised manuscript. We have made the revision as follows: “After classifying all primary gastric and metastatic ovarian samples by metastasis subtype, we further integrated the CNV and mutation data of these four groups of samples to characterize genomic changes in rich signaling pathways and compared them with GC samples in the TCGA database (Fig 4a, Supplementary Table 2).” (Line 168-171, Page 9)

Comment 11: Line 274-: Multiple testing corrections don't appear to have been used when comparing the mutation frequencies between groups. These genes may not be significant after that.

Reply: Thank you for your advice. Statistical analysis indicated that the mutations of *RHOA* (P = 0.037), *AFF2* (P = 0.028), *PIK3CD* (P = 0.028), and *TAFIL* (P = 0.028) were associated with the ineffectiveness of paclitaxel treatment, while the mutations of *SALL4* (P = 0.036), *CCDC105* (P = 0.018), and *CLDN18* (P = 0.036) were associated with the response to paclitaxel treatment (Fig. 6a). Regrettably, the exploration of genes associated with treatment response yielded no individual genes meeting an FDR < 0.05, which may be partly due to the limited sample size and the overall low mutation frequency. However, we confirmed that *CLDN18* fusion is associated with sensitivity to paclitaxel chemotherapy for ovarian metastasis in GC in *vitro* and in *vivo* (Fig. 6).

Comment 12: The Lauren classification should be provided in Table 1 if possible.

Reply: Thank you for your comments. The Lauren's classification have been provided in Table 1. We have made the revision as follows: "According to Lauren's classification, 24 patients (32.4%) were diffuse type, 21 patients (28.4%) were intestinal type, and 29 patients (39.2%) were mixed type." (Line 96-97, Page 4)

Comment 13: Line 160: GAs not Gas

Reply: Thank you for bringing this to our attention. We have made the revision as follows: "Tumor tissue samples and matched blood samples were collected for the determination of genomic alterations (GAs)." (Line 444-445, Page 21)

Comment 14: Line 183: Figure 2E not 1E

Reply: Thank you for bringing this to our attention. We have redrawn Figure 2 and checked each annotation in the revised manuscript.

Comment 15: Line 286: the meaning of “partially arising” evolution isn’t clear

Reply: Thank you for your advice. We provided a more comprehensive analysis of the shared mutations in the following figure, including the proportion of mutations in each frequently mutated gene that are present in both the primary tumor and the metastasis. This will allow for a better understanding of the mutations that are consistently present and potentially required for tumor progression and metastasis. We have made the revision as follows: “In addition, we also investigated the mutation of these genes in paired primary gastric and metastatic ovarian lesions, and found that the evolution of these genes could be classified into the following three subtypes (Supplementary Fig. 7): 1) an subtype completely arising from primary gastric lesions, such as *RHOA* mutation (6 mutations arising from primary gastric lesions/6 mutations in metastasis tumors) and *TAFIL* mutations (1/1); 2) an subtype partially arising from primary gastric lesions, such as *SALL4* (2/3), *CCDC105* (1/3), *CLDN18* (4/7), and *PIK3CD* (1/3); and 3) an subtype newly developed in metastatic lesions, such as *AFF2* mutation (0/1).” (Line 251-258, Page 11)

Comment 16: Table 2: *NTRK1* and *RAD51* aren't present in Fig 2A.

Reply: Thank you for your advice. We have added *NTRK1* and *RAD51* in revised Fig 1b. We have made the revision as follows: “Statistical analysis showed that the mutation frequencies for *FUS* ($P = 0.0004$), *ETV4* ($P = 0.0221$), *CBFB* ($P = 0.0221$), *PDGFRB* ($P = 0.0221$), *NTRK1* ($P = 0.0482$), and *RAD51* ($P = 0.0482$) were lower in metastatic ovarian lesions, compared to primary gastric lesions (Fig. 2c).” (Line 115-118, Page 5)

Reviewer #2

Comment 1: The authors mentioned that the enrichment of pathway mutations of primary GC lesions in their cohort is consistent with that of TCGA cohort. However, the DAVID/KEGG pathway analyses have not been properly presented (e.g., see Fig. 5C in <https://www.nature.com/articles/nature13480>). Can the authors present the heatmap for pathway analyses for their cohort as well as TCGA? The authors may have chosen main signaling pathways known in literature and compared them across the cohorts. However, it will be important to perform an unbiased pathway analyses to see what pathways were enriched in their cohort based on the mutations identified. Also, for Figure 3, the source for the pathways is not mentioned. The authors need to provide more details in the method section how they obtained the pathway for Fig. 3.

Reply: Thank you for providing valuable feedback on the introduction of path analysis in the manuscript.

- ① For the first question, we conducted a heat map illustrating the path analysis of four classifications (including primary, metastatic, synchronous and metachronous) and the TCGA GC database, as shown in the revised Fig. 3a. We have made the revision as follows: “After classifying all primary gastric and metastatic ovarian samples by metastasis subtype, we further integrated the CNV and mutation data of these four groups of samples to characterize genomic changes in rich signaling pathways and compared them with GC samples in the TCGA database (Fig 4a, Supplementary Table 2).” (Line 168-171, Page 8)

- ② For the second question, we conducted unbiased pathway analysis to identify the pathways enriched in our cohort, including the top5 pathways such as ERBB pathway, MAPK pathway, PI3K-AKT pathway, Wnt pathway and p53 pathway, and redrew a new Fig. 3b based on the enriched pathways and genes. We have made the revision as follows: “In primary gastric lesions, the most frequently altered genes included *TP53* (46.2%), *ERBB2* (27.7%), *CDH1* (20.0%), *ERBB3* (13.8%), and *PIK3CA* (13.8%), and were enriched in the RAP1, PI3K-AKT, MAPK and RAS pathways (Fig. 4b). For metastatic ovarian lesions, the most frequently altered genes included *TP53* (45.2%), *PIK3CA* (20.5%), *CDH1* (19.2%), *TGFBR2* (16.4%), and *ERBB2* (13.7%), and were enriched in the FOXO, P53 and ERBB pathways (Fig. 4b). The most frequently altered genes included *TP53* (44.5%), *PIK3CA* (20.5%), *ERBB2* (18.1%), *CDKN2A* (16.7%), *KRAS* (15.8%) and *APC* (14.0%) and were enriched in the ERBB, PI3K-AKT and Wnt pathways in the TCGA GC database (Fig. 4b). The majority of mutations, such as *TP53*, *ERBB2*, *ERBB3* and *TGFBR2* mutations in the MAPK pathway; *PIK3CA*, *CDH1*, *CTNBN1*, *KRAS* and *MET* mutations in the RAP1 pathway; *TP53*, *PIK3CA* and *ERBB2*, *KRAS* and *MET* in the PI3K-AKT pathway, were enriched in synchronous metastases (Fig. 4b). In metachronous metastases, the most commonly altered genes included *TP53* (57.1%), *PIK3CA* (38.1%), *CDH1* (33.3%) and *ERBB2* (23.8%), are enriched in FOXO, RAP1, PI3K-AKT and ERBB pathways (Fig. 4b).” (Line 171-186, Page 8)

Comment 2: Based on the phylogenetic analysis, the authors classified the patients into 4 categories. However, this classification seems to be just based on the number of mutations on the trunk or the branches. Can this classification be improved by fitting a mathematical tumor evolution model for each patient? For figure 4, the caption needs to be elaborated. What are the Venn diagrams representing? Are these trees corresponding to some example patients? Was there any difference in survival for these four classes?

Reply: Thank you for your valuable feedback.

① We have redrawn Figure 4 to Figure 5a, where a new classification method² is used to divide all patients into three groups: linear evolution, intermediate evolution, and parallel evolution. In the new classification system, the three groups are distinguished by genetic distance, where genetic distance > 0.7 is defined as linear, genetic distance less than 0.4 is defined as parallel, and genetic distance between 0.4 and 0.7 is defined as intermediate (Fig. 4a).

- ② The numbers in the Venn diagram represent the number of unique mutations in the primary lesion, the number of shared mutations between the primary and metastatic lesions, and the number of unique mutations in the metastatic lesion.
- ③ Each Venn plot corresponds to a real patient, and the data is sourced from the patient's paired sample sequencing results.
- ④ After categorizing the patients, we performed an efficacy assessment and conducted survival analysis on patients exhibiting varying evolutionary patterns. We have made the revision as follows: “The distinct phylogenetic patterns and genetic similarities observed in metastatic GC may have implications for clinical outcomes. Therefore, we performed an efficacy assessment and conducted survival analysis on patients exhibiting varying evolutionary patterns. Following the categorization of patients and the exclusion of samples with an insufficient number of mutations, which are not conducive to evolutionary analysis, we found that the proportion of patients who respond to paclitaxel in parallel evolution group was higher than that in linear group (72.2% vs 40.0%, $P = 0.059$), while there was no significant difference between parallel evolution group and intermediate group (72.2% vs

64.7%, $P=1$) (Fig. 5b). Survival analysis confirmed that patients in parallel evolution group had a better prognosis than those in the linear group (5-year OS: 24.93% vs 0%, $P = 0.029$), while there was no significant difference in OS between parallel evolution group and intermediate group (5-year OS: 24.93% vs 30.20%, $P = 0.981$) (Fig. 5c).” (Line 215-227, Page 10)

Comment 3: In Figure 6, the authors compared the frequency of mutations in genes involved in certain pathways. How these pathways were selected is not clear from the text. Can the authors perform unbiased pathway analysis to show how same/different pathways are enriched for synchronous vs metachronous metastasis?

Reply: Thank you for your valuable feedback. To address these issues, after classifying all primary gastric and metastatic ovarian samples by metastasis subtype, we further integrated the CNV and mutation data of these four groups of samples to characterize genomic changes in rich signaling pathways and compared them with GC samples in the TCGA database (Fig 4a, Supplementary Table 2).

Comment 4: In line no 283-289, the authors classified the evolution of genes associated with ineffective Paclitaxel therapy. However, the basis of this classification is not elaborated. The authors need to elaborate how this classification was performed and the results need to be visualized.

Reply: Thank you for your comments and suggestions on gene classification related to ineffective paclitaxel therapy in the manuscript. To address this issue, we conducted a more comprehensive analysis of the genes associated with ineffective paclitaxel treatment in the following figure, including the proportion of mutations in each frequently mutated gene present in primary and metastatic tumors. This will help to better understand the basis for this classification and the differences between different gene classes. We have made the revision as follows: “In addition, we also investigated the mutation of these genes in paired primary gastric and metastatic ovarian lesions, and found that the evolution of these genes could be classified into the following three subtypes (Supplementary Fig. 5): 1) an subtype completely arising from primary gastric lesions, such as *RHOA* mutation (6 mutations arising from primary gastric lesions/6 mutations in metastasis tumors) and *TAFIL* mutations (1/1); 2) an subtype partially arising from primary gastric lesions, such as *SALL4* (2/3), *CCDC105* (1/3), *CLDN18*

(4/7), and *PIK3CD* (1/3); and 3) an subtype newly developed in metastatic lesions, such as *AFF2* mutation (0/1).” (Line 251-258, Page 11)

Comment 5: Line no 304-312, the authors mentioned that the expression of CLDN18 was positive in some samples and negative in some other samples. It is not clear how expression can be negative. These statements need to be properly explained. The authors claim CLDN18 fusion to be a predictive biomarker for paclitaxel treatment response, however, this claim is not supported by data. While the authors show that there is no relationship between CLDN18 fusion and expression, how CLDN18 fusion is helpful in predicting the effectiveness of paclitaxel therapy is not clear. Can the authors perform any experiment to show this association?

Reply: Thank you for your questions regarding the manuscript.

- ① We detected the CLDN18 expression via IHC. Two experienced pathologists who were blinded to the clinical outcomes performed the scoring independently. CLDN18 positive was defined as moderate to strong expression in $\geq 40\%$ of tumor cells.
- ② To further clarify the correlation between *CLDN18* fusion and chemotherapy response, we constructed a stable human MKN-1 and HGC-27 GC cell lines by transfecting *CLDN18-ARHGAP26/42* lentivirus. The CCK-8 assay confirmed that *CLDN18-ARHGAP26/42* fusion can promote the ability of paclitaxel to inhibit the proliferation of GC cells, while these fusion had no significant effect on oxaliplatin inhibition of GC cell proliferation (Fig. 6c). The transwell experiments proved that *CLDN18-ARHGAP26/42* fusion can promote the ability of paclitaxel to inhibit the invasion and metastasis of GC cells, while the *CLDN18-ARHGAP* fusion had no significant effect on the invasion and migration ability of GC cells (Fig. 6d, e). Furthermore, we assessed the function of *CLDN18-ARHGAP26/42* fusion in GC ovarian metastasis mouse model. MKN-1 GC cells stably transfected with *CLDN18-ARHGAP26/42* fusion mutations or empty vector were subcutaneously inoculated into left ovary of nude mice. One week later, mice were treated with 10 mg/kg/tiw paclitaxel for 4 weeks. The results showed that the *CLDN18-ARHGAP26/42* fusion significantly increased the sensitivity of ovarian metastasis in GC to paclitaxel (Fig. 6f-h).

Comment 6: The fusion detection method is not described in detail. Since this is an important component leading one of the main findings, the authors need to elaborate the details.

Reply: Thank you for your comment regarding the fusion detection method in our manuscript. Gene fusions were detected using an in-house pipeline in our published paper³. Gene rearrangements were assessed with the Integrative Genomics Viewer.

Comment 7: The paper contains grammatical errors which need to be fixed.

Reply: Thank you for your feedback regarding the grammatical errors in our manuscript. In the revised version of the manuscript, we carefully reviewed and corrected grammatical errors to ensure the clarity and readability of the text.

Reviewer #3

Comment 1: This study lacks some wet experiments. I hope the author can add some experiments. It seems that the workload of this study is slightly small and there are some key experiments lacking.

Reply: Thank you for your questions regarding the manuscript. We demonstrated that CLDN18 fusion was associated with sensitivity of gastric cancer cells to paclitaxel therapy *in vivo* and *in vitro* as showed in Figure 6. We have made the revision as follows:

“To further clarify the correlation between CLDN18 fusion and chemotherapy response, we constructed a stable human MKN-1 and HGC-27 GC cell lines by transfecting CLDN18-ARHGAP26/42 lentivirus. The CCK-8 assay confirmed that CLDN18-ARHGAP26/42 fusion can promote the ability of paclitaxel to inhibit the proliferation of GC cells, while these fusion had no significant effect on oxaliplatin inhibition of GC cell proliferation (Fig. 6c). The transwell experiments proved that CLDN18-ARHGAP26/42 fusion can promote the ability of paclitaxel to inhibit the invasion and

metastasis of GC cells, while the CLDN18-ARHGAP fusion had no significant effect on the invasion and migration ability of GC cells (Fig. 6d, e). Furthermore, we assessed the function of CLDN18-ARHGAP26/42 fusion in GC ovarian metastasis mouse model. MKN-1 GC cells stably transfected with CLDN18-ARHGAP26/42 fusion mutations or empty vector were subcutaneously inoculated into left ovary of nude mice. One week later, mice were treated with 10 mg/kg/tiw paclitaxel for 4 weeks. The results showed that the CLDN18- ARHGAP26/42 fusion significantly increased the sensitivity of ovarian metastasis in GC to paclitaxel (Fig. 6f-h).” (Line 268-282, Page 13)

Comment 2: Can some experiments be added to illustrate the grouping of patients with partial or complete remission mentioned by the author in the method, such as immunohistochemistry?

Reply: Thank you for your comments. The response to treatment was evaluated according to the criteria of Response Evaluation Criteria in Solid Tumors (RECIST 1.1).

Patients with complete response (CR) or partial response (PR) were considered as effective, and patients with stable disease (SD) or progressive disease (PD) were considered as ineffective. H & E staining is the gold standard for chemotherapy response. Since these patients are at advanced stage when they were diagnosed, they have lost the opportunity of surgery and cannot obtain complete primary lesions for H & E staining. In clinical, the criteria of Response Evaluation Criteria in Solid Tumors (RECIST 1.1) based on CT images were always used to evaluate the treatment response of advanced GC patients. We present typical CT images of patients defined as CR, PR, SD and PD status before and after treatment as followed.

Comment 3: Line No.183, the author mentioned Fig.1E, but the submitted figures does not contain Fig.1E. I hope the author can check your manuscript carefully.

Reply: Thank you for bringing this to our attention. We have redrawn each figure and carefully checked the correspondence between the figure number and the main text. The original fig. 1E in the manuscript should be fig. 2E, and it has been adjusted to fig. 2d in the revised manuscript

Comment 4: Line No.188, the author mentioned Fig.2F, however, Figure 2 does not have an F diagram.

Reply: Thank you for bringing this to our attention. We have redrawn each figure and carefully checked the correspondence between the figure numbers and figure diagrams. The original Fig. 2F has been adjusted to the modified Fig. 2e

Comment 5: Line No.301~302 and No. 304~312, the number of samples is too small and the experimental data is insufficient to serve as strong evidence. Please improve the experiment.

Reply: Thank you for your comments. We detected the CLDN 18 expression via IHC. Two experienced pathologists who were blinded to the clinical outcomes performed the scoring independently. CLDN18 positive was defined as moderate to strong expression in $\geq 40\%$ of tumor cells. The results showed that the CLDN18 expression of primary gastric tumor was positive in 52.9% (18/34) patients in the “effective” group and 54.2% (13/24) in the “ineffective” group ($\chi^2=0.071$, $P = 0.503$, Supplementary Fig. 9a), while the CLDN18 expression of metastatic tumor was positive in 31.58% (12/38) patients in the “effective” group and 30.43% (7/23) in the “ineffective” group ($\chi^2=0.009$, $P = 0.925$, Supplementary Fig. 9b). What is more, there was no significant relationship between CLDN18 fusion and CLDN18 expression in ovarian metastasis (4/6 vs 15/55, $P = 0.130$, Supplementary Fig. 10). (Line 287-294, Page 13)

Comment 6: I hope the author will add more information about ovarian metastasis of gastric cancer in the introduction.

Reply: Thank you for your comments. We have added more information about ovarian metastasis of gastric cancer in the introduction. We have made the revision as follows:

“Gastric cancer (GC) is one of the most common malignant tumors worldwide, and metastasis and high recurrence rate are the main reasons of poor prognosis. Ovarian metastasis from GC, including synchronous metastasis and metachronous metastasis after radical surgery, accounts for approximately 5–10% of female GC patients. The prognosis of GC with ovarian metastasis is worse compared to other digestive tract-originated metastatic ovarian tumors, with a median survival time of only 8-14 months. The condition of GC with ovarian metastasis is complex, and there is still no consensus on the treatment of this disease. Currently, the systemic treatment regimens for GC patients with ovarian metastasis typically comprise chemotherapy involving agents

such as 5-fluorocrail (5-FU), platinum, or paclitaxel, but the efficacy is still unsatisfactory and it is not clear which patients could benefit from these treatments. This is also an important reason for treatment failure in female GC patients. Therefore, a comprehensive and systematic approach involving close collaboration among multiple disciplines is required to develop the most suitable individualized treatment plan for these patients.” (Line 47-61, Page 3)

Comment 7: Please explain how to determine whether the sample is gastric cancer with ovarian metastasis or primary ovarian cancer.

Reply: Thank you for your comments. According the tumor cell morphology under H & E staining of ovarian metastasis and paired primary gastric tumor, the sample is gastric cancer with ovarian metastasis or ovarian cancer can be determined. We presented typical H&E images of the primary lesion and ovarian metastatic lesion in GC patients with ovarian metastasis, and ovarian lesions in ovarian cancer patients as followed.

Comment 8: Line No.627, there are two descriptions of F in the description of Fig.2, but there is no F in Figure 2. Please check the manuscript again.

Reply: Thank you for bringing this to our attention. We have redrawn each figure and carefully checked the correspondence between the figure numbers and figure diagrams. The two descriptions of F in the description of Fig. 2 have been adjusted to the modified description e and g in revised Fig. 2

Comment 9: Can there be further in-depth validation of targeted genes? Such as adding some cellular functional experiments, etc.

Reply: Thank you for your comments. Our results indicated that CLDN18 fusion is closely related to the efficacy of paclitaxel. To further clarify the correlation between CLDN18 fusion and chemotherapy response, some functional experiments were performed *in vitro* and *in vivo*. We constructed a stable human MKN-1 and HGC-27 GC cell lines by transfecting CLDN18-ARHGAP26/42 lentivirus. The CCK-8 assay confirmed that CLDN18-ARHGAP26/42 fusion can promote the ability of paclitaxel to inhibit the proliferation of GC cells, while these fusion had no significant effect on oxaliplatin inhibition of GC cell proliferation (Fig. 6c). The transwell experiments proved that CLDN18-ARHGAP26/42 fusion can promote the ability of paclitaxel to inhibit the invasion and metastasis of GC cells, while the CLDN18-ARHGAP fusion had no significant effect on the invasion and migration ability of GC cells (Fig. 6d, e). Furthermore, we assessed the function of CLDN18-ARHGAP26/42 fusion in GC ovarian metastasis mouse model. MKN-1 GC cells stably transfected with CLDN18-ARHGAP26/42 fusion mutations or empty vector were subcutaneously inoculated into

left ovary of nude mice. One week later, mice were treated with 10 mg/kg/tw paclitaxel for 4 weeks. The results showed that the CLDN18- ARHGAP26/42 fusion significantly increased the sensitivity of ovarian metastasis in GC to paclitaxel (Fig. 6f-h).

Thank you for considering our manuscript, and we look forward to hearing from you.

Yours sincerely,

Corresponding author:

Xinagdong Cheng, MD&PhD, Professor, Chairman

1. Blokzijl F, Janssen R, van Boxtel R, Cuppen E. Mutational Patterns: comprehensive genome-wide analysis of mutational processes. *Genome Med* **10**, 33 (2018).
2. Reiter JG, *et al.* Reconstructing metastatic seeding patterns of human cancers. *Nat Commun* **8**, 14114 (2017).
3. Wu L, *et al.* Landscape of somatic alterations in large-scale solid tumors from an Asian population. *Nat Commun* **13**, 4264 (2022).

REVIEWER COMMENTS

Reviewer #1 (Remarks to the Author):

I am satisfied with the responses to earlier comments.

I have a small number of remaining questions:

Table 1 (and fig. 1b): The pathological type group is confusing. Its not clear what 'mix' is, and signet ring cell carcinoma's are still adenocarcinomas. Tumours are still called signet ring cell carcinomas even when only a minor percentage of cells are signet ring cells. Can these types be more clearly defined using a consistent nomenclature.

The top half of figure 1b would perhaps be more informative if the samples were grouped based on one of the phenotypes to the left, eg. Lauren classification.

In figure 6a, it isn't clear what the %'s refer to. What is the minus 2.4% in particular. I suggest extending the legend.

Line 329-330: some text is missing at the end of the sentence.

Reviewer #2 (Remarks to the Author):

I thank the authors for revising the paper to address the comments of the reviewers. The newly added experiments to show the correlation between CLDN18 fusion and chemotherapy response were necessary and have improved the quality of the paper. My other comments regarding the phylogeny analysis were also addressed by the authors. However, I still have some comments regarding the pathway analyses and gene classification which need to be addressed.

1. The interpretation of the results from the pathway analysis is still not unbiased and there is a lack of consistency between the pathways mentioned for the different groups and their corresponding scores in Fig. 4a. For example, for primary gastric lesions, the authors mention that mutations are enriched in the RAP1, PI3K-AKT, MAPK and RAS pathways. However, Fig. 4a shows that all these pathways have very low scores in primary gastric lesions, whereas pathways such as ERBB signaling and p53 signaling have high scores in primary gastric lesions. Same observation can be made for synchronous metastases. It is not clear how the authors are choosing the enriched pathways for each cohort. The authors mentioned that they conducted unbiased pathway analysis including top5 pathways such as ERBB pathway, MAPK pathway, PI3K-AKT pathway, Wnt pathway and p53 pathway. What is the basis of top5 pathways is not clear. The authors need to clearly mention the methodology used for unbiased pathway analysis and present the results in a more meaningful manner so that it is easy for the readers to follow. While Supplementary Table 2 shows the top 100 pathways enriched in each group, it is not consistent with Fig. 4a. Probably correct enrichment score is not being plotted in Fig. 4a.

2. Fig. 4b caption needs to be elaborated. The authors are showing frequency of mutations in genes involved in different pathways. However, it is not clearly mentioned in the caption.

3. In response to my earlier comment regarding the classification of evolution of genes associated with ineffective Paclitaxel therapy, the authors have presented a new figure that shows the mutation type for different ineffective Paclitaxel therapy associated genes and highly mutated genes in paired primary and metastasis samples. However, this figure still does not make it clear how the classification was conducted. From the figure also, three classes of genes are not observed. Is it conducted manually? Is it possible to rearrange the Supplementary Fig. 7 so that mutations belonging to the same class are present in adjacent columns? Also, for the classification, is it necessary to present all these genes in this figure? And what about genes like ERBB3 or HYDIN, which are mutated in some primary samples, some metastasis samples and in some cases, both samples. How these genes are classified?

Reviewer #4 (Remarks to the Author):

Thank you for your detailed reply to our reviewer's comments. We believe that our comments have been generally answered.

We would like to make a few additional comments on the newly added experiment in Fig. 6.

1. Lines 499-503: Please add more information on the two human gastric cancer cell lines, MKN-1 and HGC-27, including the reason for their selection in the Methods section.

2. Lines 269-270: The authors have now generated stable expression cell lines by introducing the CLDN18-ARHGA26/42 fusion gene into these two gastric cancer cell lines. Please prove that this fusion gene has actually been introduced into these cells.

3. Lines 533-534: You mention that mice were treated with paclitaxel, please describe the route of administration of paclitaxel.

That is all for additional comments.

RESPONSE TO REVIEWERS' COMMENTS

Reviewer #1:

Comment 1: Table 1 (and fig. 1b): The pathological type group is confusing. It's not clear what 'mix' is, and signet ring cell carcinomas are still adenocarcinomas. Tumors are still called signet ring cell carcinomas even when only a minor percentage of cells are signet ring cells. Can these types be more clearly defined using a consistent nomenclature?

Reply: Thank you for your valuable comments. To avoid confusion, we have now separated adenocarcinoma into two distinct categories: "non-signet ring cell" and "signet ring cell", and modified the corresponding content in Table 1, Figure 1b and revised the manuscript: "Among these patients, 62 (83.8%) were non-signet ring cell adenocarcinomas, 5 (6.7%) were signet ring cell adenocarcinomas, and 7 (9.5%) were unknown subtype (Table 1)." (Line 98-100, Page 4)

Comment 2: The top half of figure 1b would perhaps be more informative if the samples were grouped based on one of the phenotype to the left, e.g. Lauren classification.

Reply: Thank you for your valuable comments. Upon stratifying the cohort depicted in Figure 1b by age, differentiation, pathological type, metastasis subtype, extent of metastasis, and Lauren classification, we conducted comparative analyses to identify categories and genes with statistically significant differences in mutation frequencies. The results indicate that mutations in the *KMT2C* gene are more prevalent in individuals aged over 50 years as compared to those under the age of 50 ($P = 0.005$). Furthermore, when contrasted with non-signet ring cell adenocarcinoma, *CCND1* gene exhibited a higher mutation frequency in signet ring cell adenocarcinoma. Therefore, we redrawn Figure 1b and revised the manuscript "Compared to non-signet ring cell adenocarcinoma, a higher frequency of mutations in the *CCND1* genes ($P = 0.024$) was observed in signet ring cell adenocarcinoma (Fig. 1b)." (Line 110-112, Page 5)

Comment 3: In figure 6a, it isn't clear what the %'s refer to. What is the minus 2.4% in particular? I suggest extending the legend.

Reply: Thank you for your comment regarding Figure 6a. The percentages in Figure 6a are meant to represent the proportion of patients with mutations within each group. Due to our negligence, negative values should not appear here. To rectify this, we have removed the minus signs from the percentages in Figure 6a and revised the legend to accurately reflect what the percentages signify. The updated legend now clearly states: "The percentages represent the proportion of patients harboring mutations within each respective group." (Line 825-826, Page 35)

a

Comment 4: Line 329-330: some text is missing at the end of the sentence.

Reply: Thank you for pointing out the incomplete sentence at Line 329-330. Upon revisiting the manuscript, we have now completed it to accurately reflect our findings. The revised sentence is as follows: "In our cohort, comparative genomic analysis between cases of synchronous and metachronous metastasis revealed a limited number of genes with significantly distinct mutation profiles. The disparities were primarily observed in the mutation frequencies of the ERBB3 and PIK3CA genes, alongside copy number variations (CNVs) on chromosomes 5, 7, and 17." (Line 329-334, Page 15)

Reviewer #2:

Comment 1: The interpretation of the results from the pathway analysis is still not unbiased and there is a lack of consistency between the pathways mentioned for the different groups and their corresponding scores in Fig. 4a. For example, for primary gastric lesions, the authors mention that mutations are enriched in the RAP1, PI3K-AKT, MAPK and RAS pathways. However, Fig. 4a shows that all these pathways have very low scores in primary gastric lesions, whereas pathways such as ERBB signaling and p53 signaling have high scores in primary gastric lesions. Same observation can be made for synchronous metastases. It is not clear how the authors are choosing the enriched pathways for each cohort. The authors mentioned that they conducted unbiased pathway analysis including top5 pathways such as ERBB pathway, MAPK pathway, PI3K-AKT pathway, Wnt pathway and p53 pathway. What is the basis of top5 pathways is not clear. The authors need to clearly mention the methodology used for unbiased pathway analysis and present the results in a more meaningful manner so that it is easy for the readers to follow. While Supplementary Table 2 shows the top 100 pathways enriched in each group, it is not consistent with Fig. 4a. Probably correct enrichment score is not being plotted in Fig. 4a.

Reply: Thank you for your questions and comments.

- ① For each designated group—primary, metastasis, synchronous, metachronous, and TCGA cohorts—we ascertained the top 100 genes that demonstrated the highest mutation frequencies. These gene cohorts were then subjected to pathway enrichment analysis utilizing the Kyoto Encyclopedia of Genes and Genomes (KEGG) database to elucidate the significant biological pathways implicated in each group. Gene set clustering into corresponding KEGG pathways was executed via the DAVID (Database for Annotation, Visualization and Integrated Discovery) Bioinformatics Resources 6.8 platform. The outcomes of the enrichment analyses are comprehensively delineated in Supplementary Table S2. Pathways with FDR < 0.05 were defined as significantly enriched in GC.
- ② Following initial pathway categorization, we aggregated the enriched pathways from all groups to refine our focus on biological functions. Comparative pathway significance was evaluated by quantifying the prevalence of relevant genes, and a pathway score was computed for each group using the formula: $\text{Pathway Score} = (\text{Summation of mutations across genes within the pathway}) / (\text{Total gene count within the pathway} \times \text{Patient cohort size})$. Pathways were subsequently re-ranked based on these scores, with the re-ranking visually represented in Figure 4a.
- ③ After excluding some genes and pathways with high mutation redundancy, we focused on signaling pathways of P53, ERBB, Wnt, PI3K-AKT, and MAPK, which were associated with cell proliferation, migration, and differentiation. To dissect the intricate interplay and regulatory mechanisms among pivotal genes and their associated upstream and downstream mediators within these pathways, we undertook a comprehensive analysis of gene frequency and the bidirectional regulatory relationships for each constituent gene. We constructed an elaborate diagram depicting the high-frequency gene signaling networks, as delineated in Figure 4b.
- ④ For this question: “For example, for primary gastric lesions, the authors mention that mutations are enriched in the RAP1, PI3K-AKT, MAPK and RAS pathways. However, Fig. 4a shows that all these pathways have very low scores in primary gastric lesions, whereas pathways such as ERBB signaling and p53 signaling have high scores in primary gastric lesions.

Same observation can be made for synchronous metastases.” In the enrichment results of signal pathways in Supplementary table 2, Rap1, PI3K-Akt, MAPK, and Ras pathways were indeed significantly enriched in primary gastric lesions. After rescoring and clustering these signal pathways, Rap1 (score 0.29), PI3K-Akt (score 0.31), and MAPK (score 0.34) pathways still had relatively high scores in primary gastric lesions (Fig. 4a). Perhaps due to scoring 0.25 being marked as white by us, these signaling pathways appear to give a low rating feeling. To avoid misunderstandings, we have made minor adjustments to Figure 4a and supplemented the color markings in the caption of Figure 4a: “Scores ranging from 0 to 0.25 were colored to varying degrees in blue based on their size, scores 0.25 were marked as white, and scores ranging from 0.25 to 1 were marked as varying degrees in red.” (Line 804-806, Page 34)

- ⑤ To avoid misunderstandings, we have revised the description of the relevant results in the revised manuscript: “After classifying all primary gastric and metastatic ovarian samples by metastasis subtype, we further integrated mutation data of five groups of samples (primary, metastatic, synchronous, metachronous and TCGA database) for the enrichment analysis of signaling pathways in high-frequency mutated genes (Supplementary Table 2). Subsequently, for signaling pathways with FDR < 0.05, we calculated scores based on gene mutations and their frequencies within each pathway and performed clustering, ultimately presenting the signaling pathway status between different groups in Figure 4a. The results show that the p53, thyroid hormone, neurotrophin, sphingolipid, ErbB, Wnt, FoxO, AGE-RAGE, Apelin, PI3K-Akt and MAPK signaling pathways consistently achieved elevated scores across all cohorts (Fig. 4a). After excluding some genes and pathways with high mutation redundancy, we focused on signaling pathways of p53, ErbB, Wnt, PI3K-Akt, and MAPK, which were associated with cell proliferation, migration, and differentiation (Figure 4b). In the ErbB and MAPK signaling cascades, the mutational spectrum was mainly enriched in the upstream effector genes, including but not limited to *EGFR*, *ERBB2*, and *ERBB3*. The Wnt signaling pathway exhibited alterations in a series of regulatory factors, particularly involving *CTNNB1*, *APC*, *AXIN1* and *RNF43*. In the P53 and PI3K-Akt pathways, mutations were concentrated at key regulatory sites, such as *TP53*, *CDKN2A*, *CDKN2B*, *PIK3CA*, and *PTEN*, highlighting their critical roles in pathway regulation (Fig. 4b).” (Line 168-186, Page 8). We have also added a more detailed analysis process in the methods section in the revised manuscript: “For each designated group—primary, metastasis, synchronous, metachronous, and TCGA cohorts—we ascertained the top 100 genes that demonstrated the highest mutation frequencies. These gene cohorts were then subjected to pathway enrichment analysis utilizing the Kyoto Encyclopedia of Genes and Genomes (KEGG) database to elucidate the significant biological pathways implicated in each group. Gene set clustering into corresponding KEGG pathways was executed via the DAVID (Database for Annotation, Visualization and Integrated Discovery) Bioinformatics Resources 6.8 platform. Pathways with FDR < 0.05 were defined as significantly enriched in GC. Following initial pathway categorization, we aggregated the enriched pathways from all groups to refine our focus on biological functions. Comparative pathway significance was evaluated by quantifying the prevalence of relevant genes, and a pathway score was computed for each group using the formula: Pathway Score = (Summation of mutations across genes within the pathway) / (Total gene count within the pathway × Patient cohort size). Then the signaling pathways were clustered and ranked in descending order according to their respective scores. After excluding some signaling pathways with high redundancy in mutations, and focusing on

these related to cell proliferation, migration and differentiation, a chart combining gene frequency and bidirectional regulatory relationships of each gene was constructed to analyze the complex interactions and regulatory mechanisms between key genes and their related upstream and downstream mediators in these pathways." (Line 473-492, Page 22)

Comment 2: Fig. 4b caption needs to be elaborated. The authors are showing frequency of mutations in genes involved in different pathways. However, it is not clearly mentioned in the caption.

Reply: Thank you for your comment regarding the caption of Figure 4b. In response to your feedback, we have revised the caption of Figure 4b to provide a clearer explanation of what is being depicted. The updated caption now reads as follows: "Figure 4b: The distribution of mutational frequencies in the ErbB, MAPK, PI3K-Akt, Wnt and p53 signaling pathways across four groups and TCGA database. Red indicates predicted activation, blue indicates predicted inactivation. The numerical value superimposed on each box corresponds the frequency of gene mutations in the corresponding groups. The intensity of the color saturation within each box is directly proportional to the mutational frequency." (Line 806-812, Page 34)

Comment 3: In response to my earlier comment regarding the classification of evolution of genes associated with ineffective Paclitaxel therapy, the authors have presented a new figure that shows the mutation type for different ineffective Paclitaxel therapy associated genes and highly mutated genes in paired primary and metastasis samples. However, this figure still does not make it clear how the classification was conducted. From the figure also, three classes of genes are not observed. Is it conducted manually? Is it possible to rearrange the Supplementary Fig. 7 so that mutations belonging to the same class are present in adjacent columns? Also, for the classification, is it necessary to present all these genes in this figure? And what about genes like *ERBB3* or *HYDIN*, which are mutated in some primary samples, some metastasis samples and in some cases, both samples. How these genes are classified?

Reply: Thank you for your continued scrutiny and valuable feedback on our manuscript.

- ① Our findings indicate an association between seven gene variants and the therapeutic efficacy of paclitaxel, as depicted in Figure 6a. We performed a genomic profile of these genes in paired primary gastric and their corresponding metastatic ovarian lesions in revised sup figure 7. Based on their distribution patterns, these genes can be classified into three subtypes: The first subtype encompasses mutations that consistently present in both primary gastric lesions and metastatic tumors. The second subtype includes mutations that are partially present in primary gastric lesions. Lastly, the third subtype is defined by mutations are absent in primary gastric lesions and exist exclusively in the metastatic lesions
- ② To elucidate the distinctions among the tripartite classification of gene types with enhanced clarity, we have revised Supplementary Figure 7, wherein genes belonging to the same subtype are now arrayed in contiguous columns for improved comparative visualization.
- ③ Since we wanted to explore the consistency of genes related to the efficacy of paclitaxel in primary and metastatic lesions, because other genes are not related to the efficacy of paclitaxel, in order to maintain the focus of analysis, we did not demonstrate the situation of other genes in Sup Figure 7.
- ④ Considering that *ERBB3* and *HYDIN* are not related to the efficacy of paclitaxel, we did not

further demonstrate the distribution of these two genes in the primary and metastatic lesions in sup Fig7.

- ⑤ In summary, we have redrawn Sup Figure 7 and provided a new description of the results in the revised manuscript: “In addition, we conducted a comprehensive analysis of these efficacy-related genes in metastatic ovarian lesions and their corresponding primary gastric lesions. Based on their distribution patterns, these genes can be classified into three subtypes (Supplementary Fig. 7). The first subtype encompasses mutations that consistently present in both primary gastric lesions and metastatic tumors, such as *RHOA* (6 mutations in primary gastric lesions / 6 mutations in metastasis tumors) and *TAF1L* (1/1). The second subtype includes mutations that are partially present in primary gastric lesions, such as *SALL4* (2/3), *CCDC105* (1/3), *CLDN18* (4/7), and *PIK3CD* (1/3). And the third subtype is defined by mutations are absent in primary gastric lesions and exist exclusively in the metastatic lesions, such as *AFF2* mutation (0/1).” (Line 251-260, Page 11)

Reviewer #4 (Remarks to the Author):

Comment 1: Lines 499-503: Please add more information on the two human gastric cancer cell lines, MKN-1 and HGC-27, including the reason for their selection in the Methods section.

Reply: Thank you for your comments. The HGC-27 cells and MKN-1 were derived from female patients with undifferentiated adenocarcinoma. Undifferentiated adenocarcinoma exhibits strong metastatic and invasive capabilities, yet it is insensitive to chemotherapy. Our data suggest that age and the degree of differentiation are important factors affecting ovarian metastasis in gastric cancer. Hence, MKN-1 and HGC-27 were chosen for molecular biological validation. We have supplemented this section in the revised draft: “Human GC cell lines MKN-1 and HGC-27 (obtained

from Shanghai Bioleaf Biotech Co., Ltd., Shanghai, China), were derived from female GC patients with undifferentiated adenocarcinoma. These cell lines exhibited robust metastatic and invasive capabilities and demonstrated insensitivity to chemotherapy. Authentication of these cell lines was conducted using Short Tandem Repeat analysis, and regular testing was carried out to ensure the absence of mycoplasma contamination.” (Line 519-524, Page 24)

Comment 2: Lines 269-270: The authors have now generated stable expression cell lines by introducing the CLDN18-ARHGA26/42 fusion gene into these two gastric cancer cell lines. Please prove that this fusion gene has actually been introduced into these cells.

Reply: Thank you for your comments. The generated stable expression cell lines which introducing the CLDN18-ARHGA26/42 fusion gene into these two gastric cancer cell lines were confirmed by PCR detection. Quantitative PCR assays reveal a significant upregulation of the CLDN18 (exon 1-5)-ARHGAP26 (exon 7-24) fusion gene in GC cell lines after transfecting with lentivirus containing CLDN18-ARHGAP26 overexpression constructs (as shown in following Figure A-B). This is supported by markedly increased expression levels of the respective genes compared to control cohorts. The observed trend is consistently mirrored with the overexpression of the CLDN18 (exon 1-5)-ARHGAP42 (exon 7-24) fusion gene (as shown in following Figure C-D). Furthermore, validation through agarose gel electrophoresis confirms the successful expression of CLDN18-ARHGAP26/42 fusion in gastric cancer cell lines (as shown in following Figure E-F).

The qPCR primer sequences were carefully designed. For the CLDN18 (exon 1-5)-ARHGAP26 (exon 10-23) fusion gene, the forward primer sequence was GCACTGGCTTTGGGTCCAACAC, and the reverse primer sequence was CTGGCCTGTCTACTGCTCCAC. Similarly, for the CLDN18 (exon 1-5)-ARHGAP42 (exon 7-24) fusion gene, the forward primer sequence was GCACTGGCTTTGGGTCCAACAC, and the reverse primer sequence was TCCTGGGCAAGTTCATATCCCTC.

The primer sets used for PCR amplification before agarose gel electrophoresis were as follows: For the CLDN18 (exon 1-5)-ARHGAP26 (exon 10-23) fusion gene, the forward primer sequence was TTACAGCCCAACATGAACTC, and the reverse primer sequence was CCTTATAGTCCTTATCATCGTC. Similarly, for the CLDN18 (exon 1-5)-ARHGAP42 (exon 7-24) fusion gene, the forward primer sequence was TGGGCTTTCAGGACTGAAAA, and the reverse primer sequence was CCTTATAGTCCTTATCATCGTC.

Figure The generated stable expression cell lines which introducing the CLDN18-ARHGA26/42 fusion gene into HGC-27 and MKN-1 GC cell lines. A-B, Quantitative PCR assays reveal a significant upregulation of the CLDN18 (exon 1-5)-ARHGAP26 (exon 7-24) fusion gene in HGC-27 and MKN-1 GC cells after transfecting with lentivirus containing CLDN18-ARHGAP26 overexpression constructs; C-D, Quantitative PCR assays reveal a significant upregulation of the CLDN18 (exon 1-5)-ARHGAP42 (exon 7-24) fusion gene in HGC-27 and MKN-1 GC cells after transfecting with lentivirus containing CLDN18-ARHGAP42 overexpression constructs; E-F, Agarose gel electrophoresis confirms the successful expression of CLDN18-ARHGAP26/42 fusion in GC cells after transfecting with lentivirus containing CLDN18-ARHGAP26/42 overexpression constructs.

Comment 3: Lines 533-534: You mention that mice were treated with paclitaxel, please describe the route of administration of paclitaxel.

Reply: Thank you for your comments. We have described the route of administration of paclitaxel as follows: "One week later, mice were received intraperitoneal injections of paclitaxel at a dose of 10 mg/kg/tiw for 4 weeks." (Line 554-556, Page 25)

Thank you for considering our manuscript, and we look forward to hearing from you.

Yours sincerely,

Corresponding author:

Xinagdong Cheng

Xinagdong Cheng, MD&PhD, Professor

REVIEWERS' COMMENTS

Reviewer #1 (Remarks to the Author):

Thankyou for the revisions you have made.

In the integrated pathway analysis, some pathways with high mutational redundancy were excluded. This apparently included the sphingolipid pathway. Given that this pathway is so distinct to the other more classic cancer pathways, I am surprised that there was significant mutational redundancy. Given the potential for this pathway to generate novel drug targets, it would be good if more analysis could be provided.

Minor comments:

Line 40: delete 'etc'

Fig. 3b. On the axes titles, use primary and metastasis in full (not pri and meta) for clarity.

Line 192: explain the numbers - % presumably.

Line 268: ...are sensitive to...

Reviewer #2 (Remarks to the Author):

The authors have successfully addressed my comments. I am happy with the revised manuscript. I just have a minor comment.

1. In revised Supplementary Fig. 7, the color legend for Primary-metastasis does not match with the color used inside the figure. The colors should be consistent. The authors should update the color legend of this figure.

Reviewer #4 (Remarks to the Author):

I have no further questions and comments.

RESPONSE TO REVIEWERS' COMMENTS

Reviewer #1:

Comment 1: In the integrated pathway analysis, some pathways with high mutational redundancy were excluded. This apparently included the sphingolipid pathway. Given that this pathway is so distinct to the other more classic cancer pathways, I am surprised that there was significant mutational redundancy. Given the potential for this pathway to generate novel drug targets, it would be good if more analysis could be provided.

Reply: Thank you for your valuable comments. To further investigate the role of the sphingolipid pathway in gastric cancer patients with ovarian metastasis, we analyzed the drug target- related genes and key regulatory genes within the sphingolipid signaling pathway (PMID: 29147025). Our analysis showed that only 1.4% of *ABCG2* and *TERT* gene mutations were detected in metastatic lesions, and 1% of *ABCG2* and *TERT* gene mutations were detected in synchronous patients. No drug target-related gene mutations were found in the lesions of all patients (Table 1). The sphingolipid pathway exhibits a lower mutation rate in drug target-related genes and key regulatory genes, suggesting that there may be limited potential for new drug development in these patients. Consequently, this data was not presented in the revised manuscript.

Table 1 Mutation frequency of drug target genes and key regulatory genes in the sphingolipid signaling pathway in different cohorts.

Key gene in sphingolipid pathway	Gene type	The frequency of gene mutations in different subgroups (%)				
		Primary	Metastasis	Synchronous	Metachronous	TCGA
ABCC1	Regulatory genes	0	0	0	0	0
ABCG2	Regulatory genes	0	1.4	1	0	1.3
BRMS1	Regulatory genes	0	0	0	0	0
CDKN1A	Regulatory genes	0	0	0	0	1
FOS	Regulatory genes	0	0	0	0	0.2
HDAC1	Regulatory genes	0	0	0	0	1.7
HDAC2	Regulatory genes	0	0	0	0	1.5
MKRN1	Regulatory genes	0	0	0	0	0
PHB2	Regulatory genes	0	0	0	0	0

PPARG	Regulatory genes	0	0	0	0	0
PPARGC1B	Regulatory genes	0	0	0	0	0
PTPA	Regulatory genes	0	0	0	0	0
RIPK1	Regulatory genes	0	0	0	0	0.8
S1P	Drug target gene	0	0	0	0	0
S1PR1	Drug target gene	0	0	0	0	0
S1PR2	Drug target gene	0	0	0	0	0
S1PR3	Drug target gene	0	0	0	0	0
S1PR4	Drug target gene	0	0	0	0	0
S1PR5	Drug target gene	0	0	0	0	0
SET	Drug target gene	0	0	0	0	0
SPHK1	Drug target gene	0	0	0	0	1.9
SPHK2	Drug target gene	0	0	0	0	0
SPNS2	Regulatory genes	0	0	0	0	0
TERT	Regulatory genes	0	1.4	1	0	2.3
TRAF2	Regulatory genes	0	0	0	0	0.8

Comment 2: Line 40: delete 'etc'

Reply: Thank you for your valuable comments. We have deleted "etc" in revised manuscript as followed: "Additionally, by analyzing the differential response to chemotherapy, we identify potential biomarkers, including SALL4, CCDC105 and CLDN18, for predicting the efficacy of paclitaxel treatment." (Line 31-33, Page 1)

Comment 3: Fig. 3b. On the axes titles, use primary and metastasis in full (not pri and meta) for clarity.

Reply: Thank you for your valuable comments. We have redrawn figure 3b as followed.

Comment 4: Line 192: explain the numbers - % presumably.

Reply: Thank you for your valuable comments. Based on the context, we would like to emphasize the high-frequency genes in shared mutations and convert them into frequencies (%) according to your comment. In order to make this text more clear, we have made the following modifications in the revised manuscript: "The most frequently occurring mutated genes observed in shared mutations were TP53 (22/64, 34.38%), ARID1A (14/64, 21.88%), CDH1 (11/64, 17.19%), TTN (8/64, 12.50%), ERBB2 (7/64, 10.94%), and TGFBR2 (7/64, 10.94%)." (Line 185-187, Page 8)

Comment 5: Line 268: ...are sensitive to...

Reply: Thank you for your valuable comments. We have made revisions to the relevant content in the revised manuscript: "Interestingly, we found that all patients with CLDN18 fusion are sensitive to paclitaxel treatment." (Line 264, Page 12)

Reviewer #2:

Comment 1: In revised Supplementary Fig. 7, the color legend for Primary-metastasis does not match with the color used inside the figure. The colors should be consistent. The authors should update the color legend of this figure.

Reply: Thank you for your valuable comments. We updated the color legend of supplementary Figure 7 as followed.

Thank you for considering our manuscript, and we look forward to hearing from you.

Yours sincerely,

Corresponding author:

Xinagdong Cheng

Xinagdong Cheng, MD&PhD, Professor